# Regulation of heterotopic ossification by monocytes in a mouse model of aberrant wound healing

Michael Sorkin[1,8], Amanda K. Huber[1,8], Charles Hwang[1], William F. Carson IV[2], Rajasree Menon[3], John Li[1], Kaetlin Vasquez[1], Chase Pagani[1], Nicole Patel[1], Shuli Li[1], Noelle D. Visser[1], Yashar Niknafs[1], Shawn Loder[1], Melissa Scola[2], Dylan Nycz[4], Katherine Gallagher[4], Laurie K. McCauley[5], Jiajia Xu[6], Aaron W. James [6], Shailesh Agarwal[1], Stephen Kunkel[2], Yuji Mishina [7] & Benjamin Levi[1]*

Heterotopic ossification (HO) is an aberrant regenerative process with ectopic bone induction in response to musculoskeletal trauma, in which mesenchymal stem cells (MSC) differentiate into osteochondrogenic cells instead of myocytes or tenocytes. Despite frequent cases of hospitalized musculoskeletal trauma, the inflammatory responses and cell population dynamics that regulate subsequent wound healing and tissue regeneration are still unclear. Here we examine, using a mouse model of trauma-induced HO, the local microenvironment of the initial post-injury inflammatory response. Single cell transcriptome analyses identify distinct monocyte/macrophage populations at the injury site, with their dynamic changes over time elucidated using trajectory analyses. Mechanistically, transforming growth factor beta-1 (TGFβ1)-producing monocytes/macrophages are associated with HO and aberrant chondrogenic progenitor cell differentiation, while CD47-activating peptides that reduce systemic macrophage TGFβ levels and help ameliorate HO. Our data thus implicate CD47 activation as a therapeutic approach for modulating monocyte/macrophage phenotypes, MSC differentiation and HO formation during wound healing.

[1] Section of Plastic Surgery, Department of Surgery, University of Michigan, Ann Arbor, MI 48109, USA. [2] Department of Pathology, University of Michigan, Ann Arbor, MI 48109, USA. [3] Department of Computational Medicine and Bioinformatics, University of Michigan, Ann Arbor, MI 48109, USA. [4] Division of Vascular Surgery, Department of Surgery, University of Michigan, Ann Arbor, MI 48109, USA. [5] Department of Periodontics and Oral Medicine, University of Michigan, Ann Arbor, MI 48109, USA. [6] Department of Pathology, Johns Hopkins University, Baltimore, MD 21205, USA. [7] Department of Biologic and Material Sciences, School of Dentistry, University of Michigan, Ann Arbor, MI 48109, USA. [8] These authors contributed Equally: Michael Sorkin, Amanda K. Huber. *email: blevi@umich.edu

Trauma remains the number one cause of years of lost life and extremities are the most commonly injured anatomic sites[1,2]. After extremity trauma, an inflammatory response is mounted at the site of injury that promotes wound healing and regeneration of involved musculoskeletal tissues. However, severe trauma can lead to aberrant wound healing through mechanisms of dysregulated inflammation resulting in fibrosis or stem cell differentiation into a tissue distinct from its original fate, such as heterotopic ossification (HO), which commonly occurs after severe burn or trauma[3]. While the interaction between inflammatory cells and their effect on resident mesenchymal stem cells (MSCs) remains unclear, recent studies have identified MSCs capable of multi-lineage differentiation after injury[4]. The process during which HO forms is governed by MSCs that are attempting to regenerate tissue based on cues from the inflammatory niche. HO can occur in nearly any anatomic site and has been described throughout the musculoskeletal system including ligaments[5], tendon[6], periosteum[7] and the fascial or connective tissues surrounding muscles[8].

Tissue injury leads to a well-orchestrated inflammatory response with recruitment of neutrophils, monocytes and lymphocytes at different phases of inflammation. While monocytes/macrophages have been shown to play a role in aberrant wound healing,[9–11] the phenotypic and functional heterogeneity of this cell population in musculoskeletal trauma remain poorly understood. M2 macrophages have been described as reparative or anti-inflammatory and are characterized by cell surface expression of CD206 and secretion of IL-10 and TGF-β1[12,13]. Tgfb1 gene expression is thought to be specific to regenerative macrophages and regulates monocyte/macrophage function[14]. TGF-β1 is also known as a critical regulator of chondrogenic differentiation[15,16], a process that is essential in ectopic bone formation as it occurs primarily through endochondral ossification[17] and is a common feature of HO. Recent studies have implicated TGF-β1 signaling in HO[18], but its cellular origin and role on macrophage phenotype and inflammatory cell niche during aberrant bone regeneration remain unexplored.

In this study, we investigate the inflammatory response occurring at the site of musculoskeletal extremity trauma leading to aberrant cell fate causing HO. Using single cell transcriptome analyses, we identify distinct monocyte/macrophage subsets based on characteristic gene expression profiles at the early stages of inflammation and monitor subpopulation shifts using trajectory analyses. Further, we show that Tgfb1 expressing macrophages play an important role in driving aberrant mesenchymal progenitor cell differentiation leading to HO through endochondral ossification. Finally, we identify a translational strategy to modulate macrophage function by targeting cell surface receptor CD47, which not only alters Tgfb1 expression in macrophages, but also changes the phenotype of these cells, leading to attenuation of HO formation. Our findings propose a paradigm to understanding the functional impact of macrophages on MSC fate.

## Results

**Characterization of the inflammatory niche at injury site**. To understand the role of inflammation and inflammatory mediators in a post-traumatic response leading to aberrant wound healing, we analyzed both systemic and local cytokine and chemokine production by multiplex bead-based assays after a HO inducing injury. Burn tenotomy (burn/tenotomy) was performed on wild type mice, and tissue homogenates from the extremity injury site, and plasma were collected at days 0 (no burn/tenotomy), 3, and 7 post burn/tenotomy. Multiplex protein analysis revealed local increases in monocyte and neutrophil associated cyto/chemokines. In particular, chemokines responsible for recruitment and

activation of these cells were increased at the tenotomy site at 3 days post injury, including CXCL1, CXCL2, and CCL2 (MCP-1) (Fig. 1a). Additionally, monocyte produced chemokines CCL3 and CCL4 were increased (Fig. 1a). CCL3 and CCL4 have been shown to induce the expression of other pro-inflammatory molecules including IL-1, IL-6, and TNF-α[19], which were also increased at the site of extremity injury in our model (Fig. 1b). Cytokines G-CSF and GM-CSF, important in neutrophil and monocyte/macrophage maturation respectively, were also increased (Fig. 1c). These data point towards neutrophils and monocytes being important players of the immune system during the initial response to musculoskeletal trauma. In addition to neutrophil and monocyte associated factors being increased locally at the site of injury, transforming growth factor beta (TGF-β) 1, 2, and 3 were also increased (Fig. 1d). Interestingly, LIF and CXCL5 were increased, which, in addition to immune modulatory functions, are believed to be important in stem cell recruitment and maintenance (Fig. 1e). Systemically, there were less changes in these inflammatory mediators in the plasma from days 0, 3, and 7 post burn/tenotomy with no significant fluctuations identified over the time course of the experiment (Supplemental Fig. 1). This suggests that changes in monocyte and granulocyte associated factors may be important in the local microenvironment leading to aberrant tissue regeneration as seen in HO formation.

To obtain an unbiased characterization of cells infiltrating the area of injury and producing these cyto/chemokines, we harvested the tissue at the tenotomy site at day 3, isolated, and performed single cell RNA sequencing using the 10× genomics platform. Cluster analysis using the t-distributed stochastic neighbor embedding (tSNE) dimensionality reduction identified that there were 14 different cell clusters present at the day 3 injury site. Using the Cten and Immgen databases, we identified 5 monocyte/macrophage clusters and two granulocyte clusters based on their top ~50 gene expressing signatures (Fig. 1f). Using feature plots to identify the expression of the above immune mediators; we found that the monocyte and macrophage clusters were responsible for much of this expression. The stromal cells also appeared to contribute to recruitment, as these cells expressed Cxcl1, Cxcl2, and Ccl2 (Fig. 1g). Tgfb1 was the only TGF beta family member expressed in the monocyte and macrophage clusters (Fig. 1g). Clusters 0, 1, and 3 represent the clusters with most of the Tgfb1 gene expression. Analyzing the top genes specific in these clusters and doing a literature search of other single cell analysis, we find that populations similar to those seen in our clusters 1 and 3 have been described (Supplemental Table 1). Gene signature similar to our cluster 1 has been seen in atherosclerosis[20] and acute lung injury[21] while signatures similar to cluster 3 were seen in myocardial infarction[22] and atherosclerosis[20]. Cluster 0 does appear unique, as this cluster appears similar to a signature seen when analyzing monocyte-derived clusters from the central nervous system under homeostasis[23]. Further, to understand the cells responding to these mediators, analysis of respective receptors was performed. As would be expected, we found that granulocyte colony stimulating factor receptor, Csf3r, and the neutrophil chemokine Cxcl2 receptor (Cxcr2) were expressed in the granulocyte clusters. The macrophage colony stimulating factor receptor, Csf1r, responsible for binding M-CSF leading to survival, proliferation, and differentiation of macrophages[24], was expressed in the macrophage clusters (Fig. 1h). Unexpectedly, Csf1r expression was also seen in our lymphocyte and granulocyte clusters. Previous studies have shown that while the protein is not produced, transcripts for Csf1r can be seen in neutrophils, dendritic cells, and splenic T cells[25–27]. Together these data highlight a key role for monocytes, macrophages, and granulocytes in the initial phases of the pathogenesis of HO.

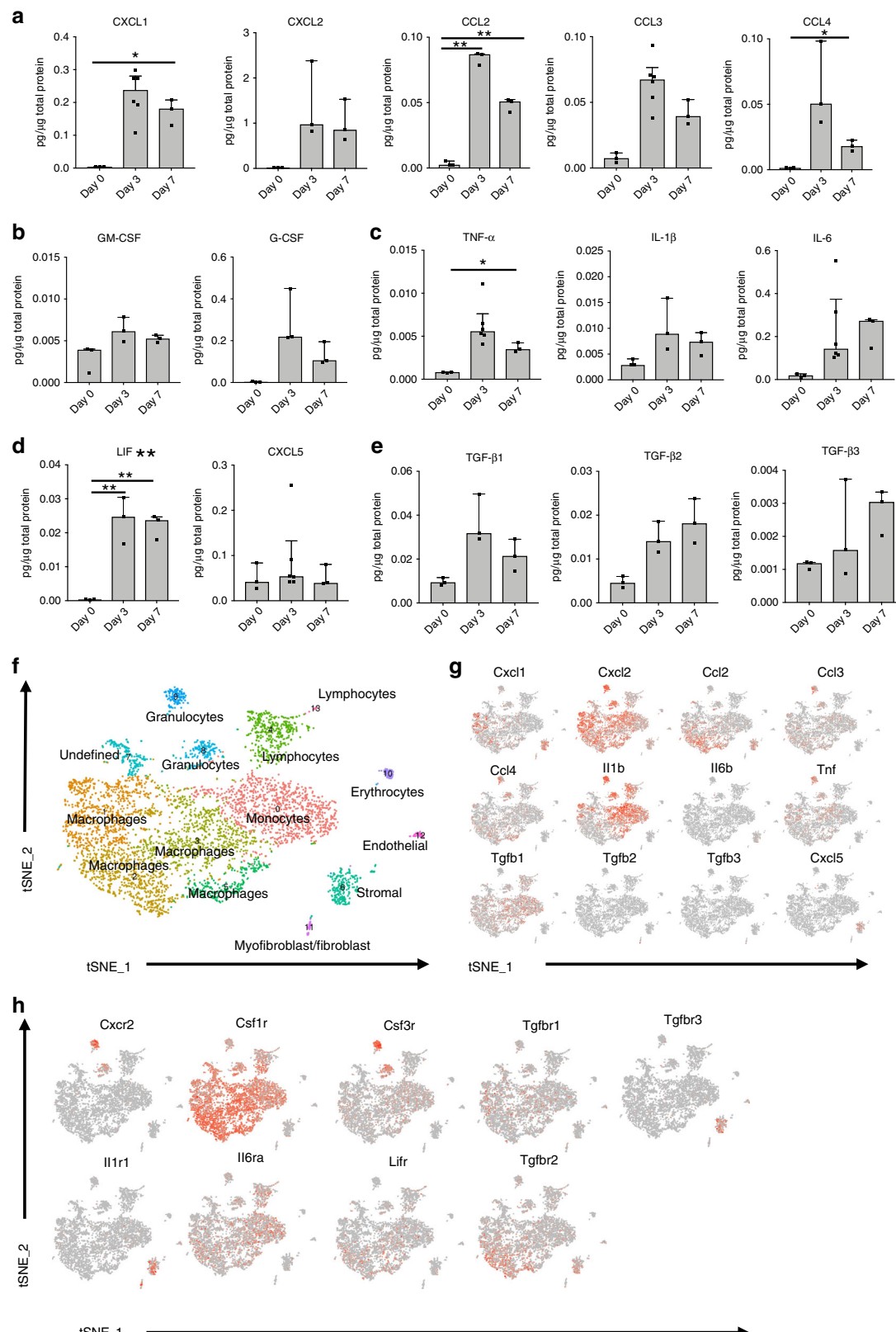

**Critical initial functions of monocyte-derived macrophages**. Since monocyte and macrophage clusters compose the majority of the immune cells present at day 3 in our model and appear to be the cells producing mediators increased at this time point, we performed in depth in vivo analysis of the cellular composition

and function of infiltrating immune cells over the course of HO formation. To do this, we first monitored the recruitment of myeloid cells to the site of injury using a reporter mouse model driven by myeloid lineage specific protein LysM (LysM-Cre/mTmG$^{fl/fl}$). Using these mice, we found that green fluorescent

**Fig. 1 Characterization of the inflammatory niche and immune cell infiltrate at the site of the extremity injury reveals a role for monocytes and macrophages in the initial phases of the pathogenesis of HO. a–c** Injury site homogenates harvested from burn/tenotomy mice on day 0, 3, and 7 post burn/tenotomy. **a** Monocyte/Macrophage associated factors. **b** Monocyte/Macrophage and neutrophil maturation factors. **c** Cytokines stimulated by monocyte factors. **d** TGF family members. **e** Stem cell maintaining factors. Levels of cytokines and chemokines in pg/ug of total protein, data represented as the median with interquartile range. Changes in cytokines and chemokines across day 3 and day 7 vs. day 0 were analyzed by an analysis of variance (ANOVA) with post-hoc Dunnett test ($n = 3$ mice/time point) significance. Non-heteroscedastic data identified by Levene's test for homogeneity of variances were alternatively analyzed by Welch statistic and post-hoc Dunnett T3. Degrees of freedom (df or df1) across samples = 2. F statistic and significant post-hoc $p$-values respectively: CXCL1: 30.359, $p$(D0 vs. D7) = 0.036, CXCL2: 8.504, CCL2: 268.773, $p$(D0 vs. D3) = 0.000, $p$(D0 vs. D7) = 0.000, CCL3: 16.430, CCL4: 22.441, $p$(D0 vs. D3) = 0.014, G-CSF: 12.579, GM-CSF: 4.988, IL-1b: 3.486, IL-6: 13.019, TNF-α: 38.435, $p$(D0 vs. D7) = 0.019, TGF-β1: 9.156, TGF-β2: 11.376, TGF-β3: 7.362, CCL5: 0.825, CXCL5: 0.825, LIF: 25.368, $p$(D0 vs. D3) = 0.001, $p$(D0 vs. D7) = 0.002 *$p$ < .05 **$p$ < .01. **f** t-SNE dimensionality reduction analysis of single cell sequencing from day 3 cells harvested at the extremity injury site revealed 14 distinct cell clusters (representative, performed in triplicate). **g, h** Feature plots displaying the single cell gene expression of **g** monocyte/macrophage cytokines and chemokines increased in the homogenates and **h** their receptors. Source data are provided as a Source Data file.

myeloid cells were present in high numbers at the injury site 1 week after injury and remained present as late as 3 weeks after injury (Fig. 2a). Next, we monitored myeloperoxidase (MPO) activity, an enzyme commonly expressed in activated neutrophils, monocytes, and macrophages, at the injured site using bioluminescent in vivo imaging over a course of 3 weeks. MPO activity was detected specifically at the ankle and reached a peak on day 1 after injury (Fig. 2b) consistent with an influx of neutrophils observed by flow cytometry at the same time point (Fig. 2c, d). MPO activity decreased thereafter but remained detectable even 3 weeks post-trauma at the tendon injury site, after neutrophils were no longer present, indicating that MPO is derived from recruited monocytes/macrophages at later time points (Fig. 2b, d). In contrast, MPO activity almost entirely disappeared at the burn site on the back where no HO formed (Fig. 2b).

Using flow cytometry, we observed an early infiltration of CD11b+Ly6G+ neutrophils that peaked at day 1 comprising 19% of all cells at the injury site (Fig. 2d, negative gates Supplemental Fig. 2). Similarly, CD11b+Ly6G− monocytes were recruited in large numbers comprising over 40% of all cells at day 2. While neutrophils mostly disappeared after 3 weeks, monocytes/macrophages remained the prevalent inflammatory cell population comprising nearly 15% of all cells by 3 weeks (Fig. 2d). Further, we noted that Ly6Chi monocytes, commonly described as the classical pro-inflammatory monocytes[28], were recruited during the initial stage of the early inflammatory phase and rapidly dissipated by day 5, while Ly6Clo macrophages were present later in the inflammatory response and were the prevalent population that persisted out to 3 weeks after injury (Fig. 2d). Ly6Clo cells also co-expressed the markers F4/80 and CD206 suggesting a more M2 like phenotype (Fig. 2d).

Taken together, this data suggests that monocyte-derived macrophages are important in the initial immune response to musculoskeletal injury, and responses by these cells after injury in this model might contribute to aberrant tissue regeneration as seen in HO.

**Macrophage depletion attenuates aberrant wound healing**. Having shown that monocytes are recruited to the site of injury in large numbers we aimed to understand their role in promoting HO formation by using intravenous liposomal clodronate depletion of circulatory monocytes prior to and immediately following burn/tenotomy (Fig. 3a). Clodronate liposomes injected intravenously cannot cross capillary barriers[29] and therefore do not affect resident macrophages[30] allowing us to validate the role of primarily circulatory monocytes. Depletion of monocytes was confirmed using flow cytometry, demonstrating significant reduction of the circulatory monocyte population (6.5% vs. 1.1%, $p < 0.001$) at one week after injury without depletion of neutrophils (Fig. 3b). Administration of clodronate for the first three

weeks after injury resulted in a significantly dampened inflammatory response at the injury site, reinforced by decreased MPO activity (Fig. 3c) and reduced ankle edema size (Fig. 3d), confirming a critical role of infiltrating monocytes in mounting inflammation. Correspondingly, depletion of circulatory monocytes resulted in decreased recruitment of total monocytes to the site of injury while neutrophil recruitment remained unaffected (Fig. 3b). Interestingly, the decrease in monocytes was attributable to a significant reduction of Ly6Clo monocytes; however, the percentage of Ly6Chi monocytes remained unchanged. Consistent with these findings, the presence of differentiated mature macrophages (CD11b+F4/80+) were significantly reduced (18.0% vs. 10.7%, $p < 0.001$) (Fig. 3b). To determine changes in the inflammatory milieu both locally and systemically after total monocyte depletion, we used LysMcre-iDTR mice where monocytes were depleted by pre-injection of diptheria toxin (DT) two days before the burn/tenotomy, the day of burn/tenotomy and 2 days after the burn/tenotomy. demonstrated changes to the pro-inflammatory cyto/chemokine levels in the plasma at day 3 (Supplemental Fig. 3). We found changes with regards to pro-inflammatory cyto/chemokines are in the serum at day 3. This suggests that the depletion of macrophages affects the systemic response to the dorsal burn but does not change the early response occurring at the HO site. We found a trend for decreases in plasma CCL2, G-CSF, GM-CSF, IL-1b, IL-6, and TNF-α levels when macrophages were depleted. There were also decreased levels of TGF-β1 at the tenotomy site, albeit not significant (Supplemental Fig. 3). This suggests that depletion of monocytes alters the peripheral response after injury, subsequently affecting the downstream immune cell response which occurs at the injury site.

To determine whether the suppression of the early inflammatory response through monocyte depletion affects pathologic wound healing, we examined the effect on early formation of HO. Mice treated with liposomal clodronate demonstrated decreased formation of the cartilage precursor as evidenced in Safranin O staining of the injury site at 3 weeks (Fig. 3e). As expected, early depletion of circulatory monocytes further resulted in reduced volume of mature HO by microCT quantification in the clodronate group as compared to control (Fig. 3f). These findings reveal a critical role of monocytes/macrophages in initiating the acute inflammatory phase after tissue injury confirming a contribution to aberrant mesenchymal cell differentiation.

**Monocyte/macrophage heterogeneity during trauma induced HO.** To appreciate the changes of monocytes/macrophages during the initial phases leading to HO formation in an unbiased approach, single cell RNA (scRNA) sequencing was performed from tissue at the extremity injury site collected at days 0, 3, 7, and 21-post injury. Canonical correlation analysis of the

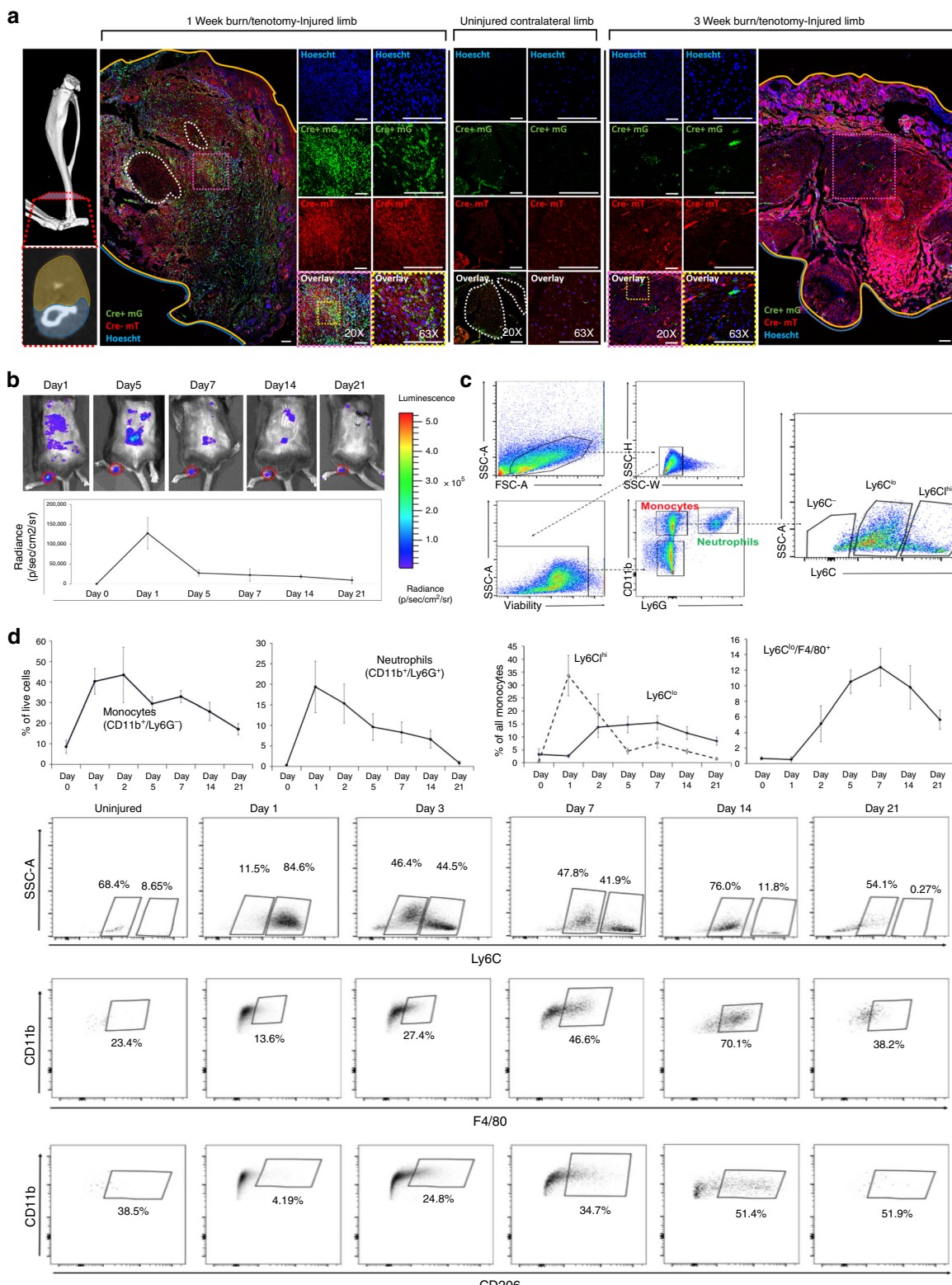

**Fig. 2 Monocyte-derived macrophages are important in the initial immune response to musculoskeletal injury and persist 3 weeks after injury.**
**a** Immunofluorescence of LysmCre$^{+/-}$/mTmG$^{fl/fl}$ mice at 1 and 3 weeks after injury localizing EGFP+ myeloid cells. Scale bars correspond to 100 μm. **b** In vivo measurement of inflammation with imaging of myeloperoxidase (MPO) activity. $n = 3$ mice per indicated time point. **Top:** representative images of each time point (scale shown: blue = low, red = high). **Bottom:** quantification of the total bioluminescent signal at the injury site using standardized region of interest (ROIs). **c** Gating strategy for flow cytometry analysis to identify inflammatory cell populations. **d Left:** Quantification of recruitment of inflammatory cell populations over time was analyzed using flow cytometry ($n = 4$ mice per time point). **Right:** Representative flow cytometry plots demonstrating dynamic changes in Ly6C, F4/80 and CD206 monocyte populations over 3 weeks ($n = 4$ mice per time point). Source data are provided as a Source Data file.

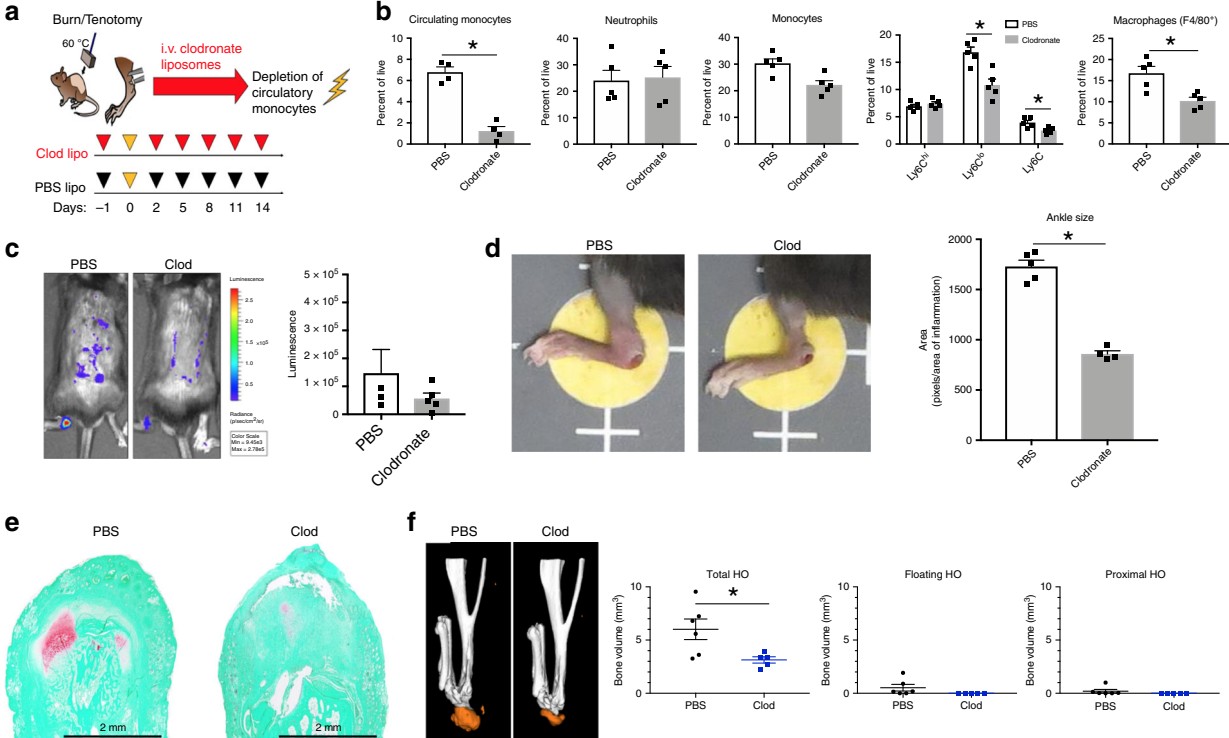

**Fig. 3 Macrophage depletion reduces acute inflammation and aberrant musculoskeletal wound healing. a** Schematic of experimental set-up. **b** Flow cytometry analysis of injury site 1 week after burn/tenotomy in mice treated with clodronate or PBS for monocytes (CD11b$^+$ Ly6G$^-$); neutrophils (CD11b $^+$Ly6G$^+$); classical monocytes (CD11b$^+$ Ly6G$^-$ Ly6C$^{hi}$), alternatively activated monocytes (CD11b$^+$ Ly6G$^-$ Ly6C$^{lo}$) and macrophages (F4/80$^+$). Circulating monocytes: $n = 4$/group, df $= 8$, $t = -0.213$, $p = 0.000$. Injury site: $n = 4$/group, df $= 8$. Neutrophils: $t = -0.213$, $p = 0.837$; Monocytes: $t = 3.490$, $p = 0.008$; Ly6C$^-$: $t = 3.193$, $p = 0.013$; Ly6C$^{low}$: $t = 4.139$, $p = 0.003$; Ly6C$^{hi}$: $t = -0.979$, $p = 0.356$; F4/80 macrophages: $t = 3.552$, $p = 0.007$. **c** IVIS imaging of MPO activity 1 week after injury in mice treated with clodronate or control ($n = 4$ mice per treatment). Total bioluminescent signal at the injury site using standardized region of interest (ROIs) was calculated and presented as total flux in photons per second per ROI. df $= 3.182$, $t = 1.995$, $p = 0.135$. **d Left:** representative images of ankle edema present in each treatment group. **Right:** quantification of ankle size. $n$(PBS) $= 5$, $n$(Clod) $= 4$, df $= 7$, $t = 11.350$, $p = 0.000$. **e** Representative Safranin O staining of tendon injury site 3 weeks after burn/tenotomy in clodronate and PBS treated mice. $n = 3$/group. **f** MicroCT analysis of tenotomy site 9 weeks after burn/tenotomy in clodronate and PBS control treated mice. **Left:** representative 3D reconstruction. **Right:** quantification of unthresholded total HO, floating HO (HO not associated with tibia or calcaneus) and proximal HO (HO proximal to the calcaneus). $n$(PBS) $= 6$, $n$(Clod) $= 5$. Total HO: $t = 3.302$, df $= 5.312$, $p = 0.020$; Floating HO: $t = 1.867$, df $= 5.002$, $p = 0.121$; Proximal HO: $t = 1.313$, df $= 9$, $p = 0.222$. All analyses assess for homoscedasticity and difference in means via Levene's F-test and two-tailed Student's t-test, respectively. *$p <$ 0.05. Source data are provided as a Source Data file.

4 datasets yielded 12 transcriptionally unique cell clusters identifiable at the injury site, several with characteristic profiles attributable to known cell types including those classified as stromal cells by expression of *Pdgfrα* (clusters 0 and 6) and a large number of cells expressing markers being related to granulocytes and monocytes/macrophages (clusters 1, 3, 4, 7; Fig. 4a). Distinct clusters expressed genes commonly linked to M2 monocytes including *Mrc1* (CD206) and *Arg1* (cluster 1, 3, and 4; Fig. 4b). These findings confirm cellular heterogeneity among monocytes and macrophages at the site of injury during trauma induced HO. Analysis of cells from day 0 (prior to injury) in the canonical analysis (Fig. 4c) demonstrated for the first time in this model, macrophages that were present at steady state (resident macrophages) that may play a role in the pathogenesis and the formation of HO (Fig. 4c; cluster 1). To determine markers for this population, we analyzed feature plots from day 0 tSNE plots. Resident macrophages appear to be defined as *Siglec1*$^+$, *Timd4*$^+$, *Lyve1*$^+$, *Cd163*$^+$, *Arg1*$^-$, *Fcgr1*$^+$ cells (Fig. 4d).

To understand the hypothetical developmental relationships that might exist within the monocyte and macrophage clusters, we performed trajectory analysis on clusters 1, 3, 4, and 7 using the Monocle algorithm. Three branch points were determined based on changes in monocytes and macrophages gene

expression, and this was plotted in pseudotime (Fig. 5a). Clusters were superimposed on the monocle pseudotime plot and revealed that cluster 1 fell towards the beginning of pseudotime while clusters 4, 3, and 7 in that order, followed along the remaining trajectory (Fig. 5b). Gene expression plotted by cluster across pseudotime, revealed that increased expression of *Arg1* was accompanied by a decreased expression of *Siglec1, Mrc1,* and *Csfr1* (Fig. 5c). Analysis of gene expression changes over pseudotime at each branch point (Fig. 5d) allowed us to determine cell markers associated with each branch of the trajectory. Cells at the beginning of the trajectory are similar to our resident population having markers S*iglec1, Mrc1, Folr2,* and are *Arg1* negative (Fig. 5d; left at midline). The most up-regulated markers for each trajectory are listed in Fig. 5a. Tracking gene expression changes across macrophage states revealed coordinated patterns defining macrophage subset identification and functions. In short, changes after the first branch resulted in two populations: one that was *IL-1b*$^+$ *Ccl4*$^+$, *Erg1*$^+$, *Mrc1*$^-$, *Siglec1*$^-$, and *Csfr1*$^-$, while the other was *Mfge8*$^+$, *Mgp*$^+$, *Siglec1*$^+$, *Arg1*$^+$, and *Cd68*$^+$. After the second branch there was a terminal *Siglec1* $^+$, *Arg1*$^+$, *Cd68*$^+$, *Tgfb1*$^+$, *Clec4d*$^+$, *Tnf*$^+$, *Cxcl3*$^+$, *Cd14*$^+$, and *Chil3*$^+$ signature and an intermediate *Mmp2*$^+$, *Cd81*$^+$, *Sparc*$^+$, and *Igfbp7*$^+$ subset. At the final branch, there were two subsets

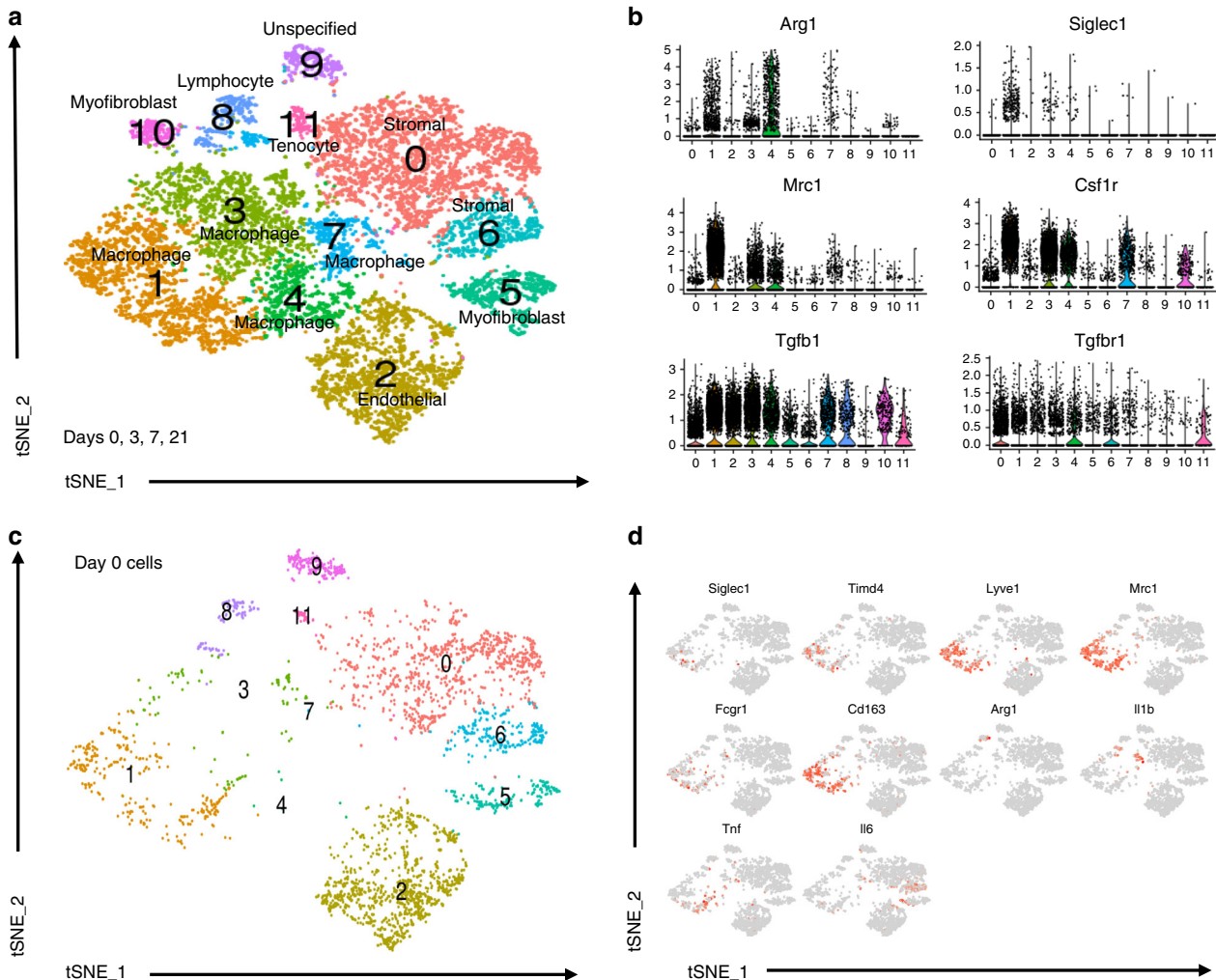

**Fig. 4 Single cell RNA sequencing reveal multiple monocyte and macrophage clusters during trauma induced HO. a** Day 0, 3, 7, and 21 combined canonical correlation analysis and T-distributed stochastic neighbor embedding (t-SNE) plot identified 12 distinct cell clusters based on gene expression differences. **b** Violin plots of monocyte/macrophage markers (Siglec1, Arg1, Mrc1, Csfr1), Tgfb1, and Tgfbr1. **c** t-SNE plot displaying only those cells in the canonical correlation analysis from day 0. **d** Feature plots of monocyte/macrophage genes expressed to identify possible resident macrophages at the extremity injury site from day 0 cells.

characterized by *Trem2*[+], *Apoe*[+], *Spp1*[+], *Arg1*[+], and *Tgfb1*[+] expression in the first, and *Plac8*[+], *Ifitm6*[+], *Igfbp4*[+], and interestingly, *Col1a1*[+] *Col1a2*[+] *and Col3a1*[+] expression in the other (Fig. 5a). Taken together this highlights the heterogeneity that is present in the macrophages at the site of extremity injury during trauma-induced HO.

***Tgfb1* expression in macrophages drives HO formation**. Having established that monocytes and monocyte-derived macrophages are important for HO formation in our burn/tenotomy model, we further explored the pathways underlying this process. Microarray gene set enrichment analysis of peripheral blood mononuclear cells from patients who suffered burn injuries, predisposing them to a higher risk of HO formation, demonstrated elevated TGF-β1 signaling (Fig. 6a; left). Subsequently, HO anlagen was harvested from injured wild type mice and subjected to RNAseq analysis. Interrogation of the data using a publicly accessible gene set enrichment database revealed clustering of highly expressed *Tgfb1* signature genes indicating an activation of the TGF-β1 pathway at the injury site (Fig. 6a; right). Furthermore, we observed increased pSMAD2/3 in tissue

homogenates of injured sites by 5 days after injury corresponding to the peak of monocyte recruitment (Fig. 6b). Further, TGF-β1 co-localized with F4/80+ and CD68+ macrophages at the early HO site in both mouse (Fig. 6c) and human (Fig. 6d; left) samples, respectively. In early human HO, pSMAD2 co-localized with PDGFRα, an MSC marker, suggesting TGF-β1 signaling present in these cells at the HO site (Fig. 6d; right). Homogenates from the day 3 extremity after burn/tenotomy displayed high levels of TGF-β1 (Fig. 1d) and single cell analysis from day 3 post injury demonstrated that *Tgfb1*, but not *Tgfb2* or *Tgfb3*, was increased within the monocyte and macrophage clusters (Fig. 1g). Finally, in our single cell and trajectory analysis *Tgfb1* is up-regulated in many of the macrophage cell clusters at the HO site (Fig. 5a). Together, these data suggest that *Tgfb1* expressing monocytes/macrophages during the early stages of inflammation, may be important in the aberrant formation of HO in human and mouse.

To confirm this finding, we hypothesized that monocyte *Tgfb1* deletion would result in attenuation of HO formation. To test this, we crossed LysMCre mice with *Tgfb1*fl/fl mice, creating a myeloid cell specific deletion of *Tgfb1*. These mice did not have differences in body weight or defects in cortical bone thickness (Supplemental Fig. 4) and had demonstrated decreased TGF-β1

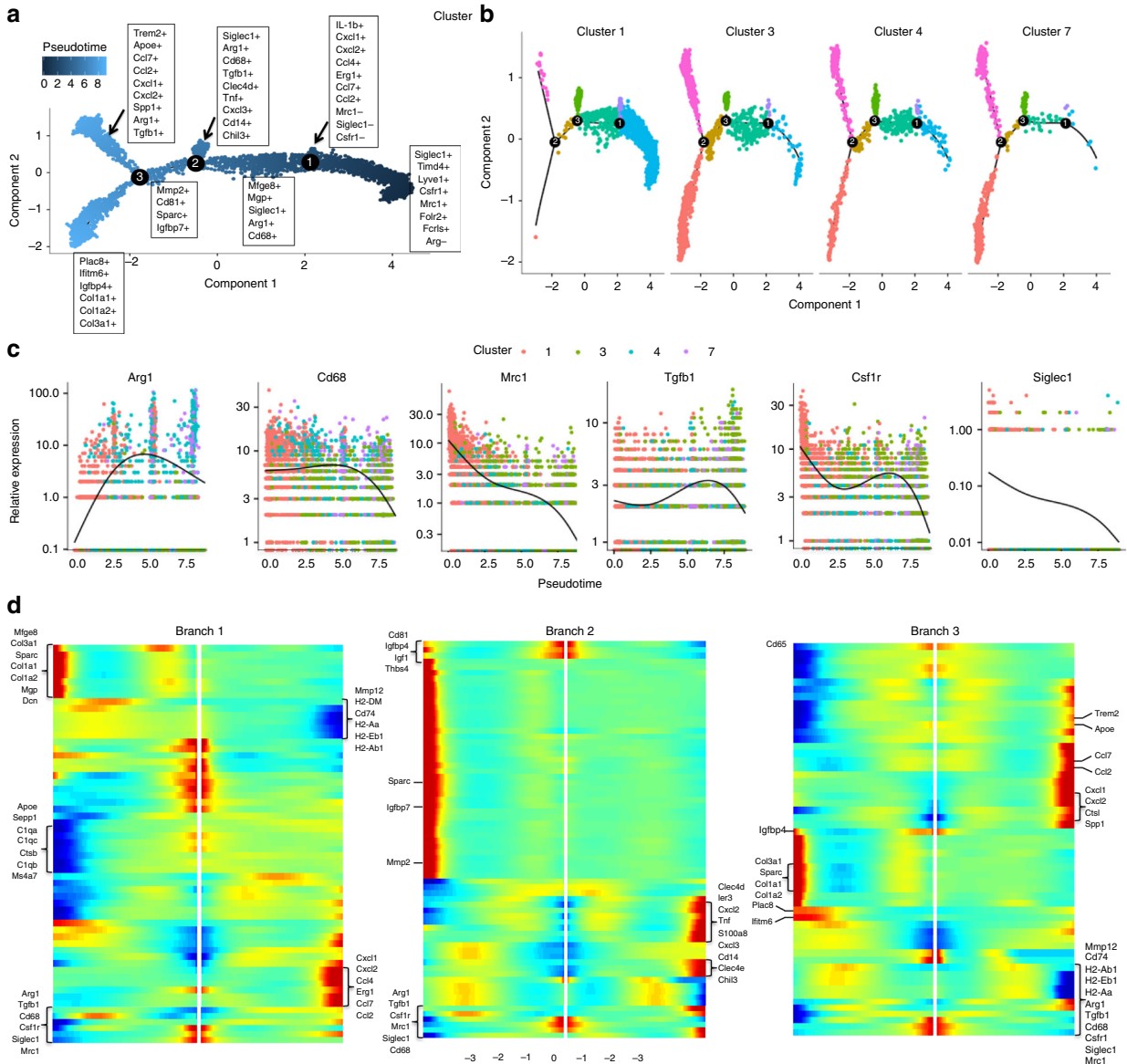

**Fig. 5 Trajectory analysis of monocyte and macrophages from scRNA occupy differential activation states. a** Focused Monocle pseudotime trajectory analysis including only the monocyte/macrophage defined clusters 1, 3, 4, and 7 from Fig. 4a. **b** Monocyte/ macrophage clusters superimposed on pseudotime branches. **c** Cluster-defined gene expression plotted as a function of pseudotime. **d** Heatmaps of differentially expressed genes ordered based on their common kinetics through pseudotime displayed at each trajectory branch point as defined in our Monocle trajectory analysis of the monocyte/ macrophage clusters (5a).

expression in bone marrow derived macrophages (Supplemental Fig. 5). LysMCre-*Tgfb1*^fl/fl^ (m*Tgfb1*KO) mice demonstrated a significantly decreased inflammatory response after burn/tenotomy as evidenced in lower MPO activation at the ankle and on the back indicating that *Tgfb1* critically regulates inflammatory function of the infiltrating monocytes in the acute injury (Fig. 7a). Given that *Tgfb1* can alter monocyte chemotaxis[31], we next interrogated whether inflammatory cell recruitment was altered in m*Tgfb1*KO mice after an HO inducing injury. Seven days after injury, deletion of *Tgfb1* in myeloid lineage cells did not alter the recruitment of cells to the site of injury (Fig. 7b). Most interesting, m*Tgfb1*KO mice demonstrated significantly reduced HO formation at 9 weeks after injury by microCT (Fig. 7c). Analyzing TGF-β1 levels in homogenates and serum from m*Tgfb1*KO mice after burn/tenotomy, we observed no decrease in TGF-β1 at the injury

site, but a trend towards a decrease in the serum of m*Tgfb1*KO mice compared to wild type (Fig. 7d), albeit not significant. These data point to *Tgfb1* being deleted from myeloid lineage cells in these mice (lower levels in serum); however we still identified high levels at the site of injury, suggesting that it is not the change in the microenvironment, but the loss of TGF-β1 resulting in an altered macrophage phenotype and functional capacity to promote HO at the site of injury. Analysis of the plasma and tenotomy site in LysMCre-*Tgfb1*^fl/fl^ mice was also performed. There were no significant changes in the levels of pro-inflammatory cyto/chemokines at the tenotomy site. There was a trend for decreases in plasma GM-CSF, IL-1b, IL-6, CCL3, CCL4, CCL5, but no differences in the levels of TGF-β1 (Supplemental Fig. 6). This suggests depletion of TGF-β1 producing monocytes alter the peripheral response and

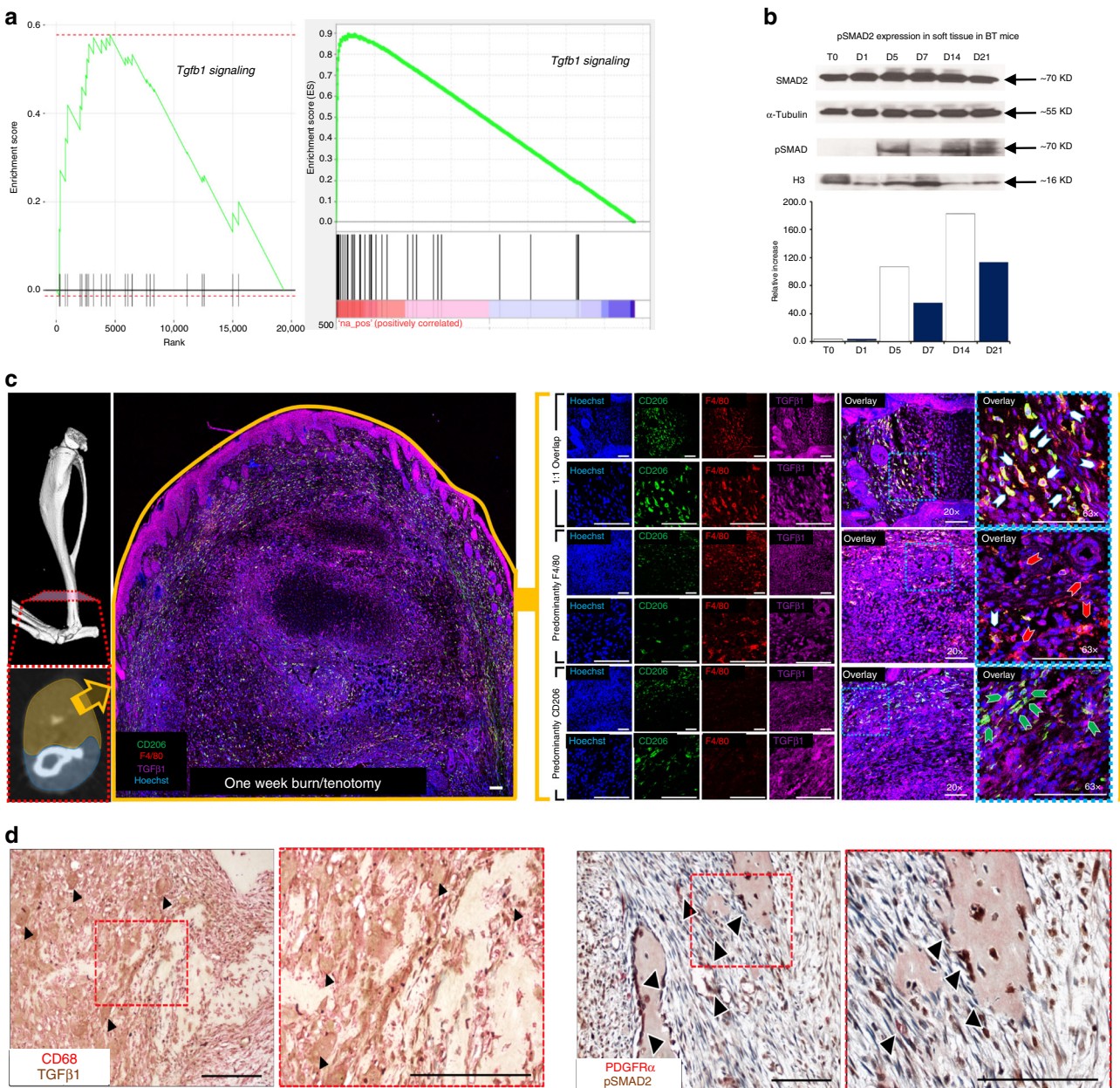

**Fig. 6 TGF-β1 expressing macrophages are present during HO formation. a Left:** GSEA analysis of microarray data collected from buffy coat of human burn injury patients at increased risk of HO compared to post-surgical control patients. **Right:** GSEA analysis of RNAseq performed of tendon injury site 3 weeks after burn/tenotomy in mice. **b** Western blot of whole tissue protein collected from the injury site of C57BL/6 J mice after burn/tenotomy at indicated time points. A western blot for p-SMAD2 and H3 was performed on the nuclear fraction and SMAD2 and alpha-Tubulin were performed on the cytosolic fraction. *n* = 5 were pooled for each time point. **c** Co-localization of F4/80+ and TGF-β1 at tendon injury site 1 week after burn/tenotomy. Scale bars correspond to 100 μm. **d Left:** Co-localization of CD68+ and TGF-β1 in early human HO. **Right:** Co-localization of p-SMAD2 and PDGFRα in human HO. Source data are provided as a Source Data file.

subsequent response occurring at the injury site. This data points to the TGF-β1 production in monocytes as a marker of the cell phenotype important in potentiating ectopic bone formation.

**CD47-activating peptide alters macrophage phenotype.** Given that complete macrophage ablation would abrogate phagocytosis and oxidative burst capacity of macrophages necessary for normal wound healing, and because TGF-β1 signaling plays a critical role in the homeostasis of several essential physiologic processes, global systemic depletion of macrophages could have considerable side effects. Because of this, we sought to find a more

clinically applicable approach to altering macrophage phenotype. Previous studies have suggested that CD47 activation modulates *Tgfb1* expression[32,33], therefore, we treated mice that underwent burn/tenotomy with daily systemic injections of CD47-activating peptide (p7N3) for 3 weeks. Treatment with CD47-activating peptide lead to significantly decreased cartilage formation indicated by Safranin O staining (Fig. 8a) as well as decreased mature HO formation seen by microCT (Fig. 8b). CD47 treatment did not impact the cortical thickness of the tibia (Supplemental Fig. 7). As was seen with m*Tgfb1*KO mice, local TGF-β1 protein levels were not different with CD47 treatment, however there was a decrease in the systemic levels of circulating TGF-β1 (Fig. 8c).

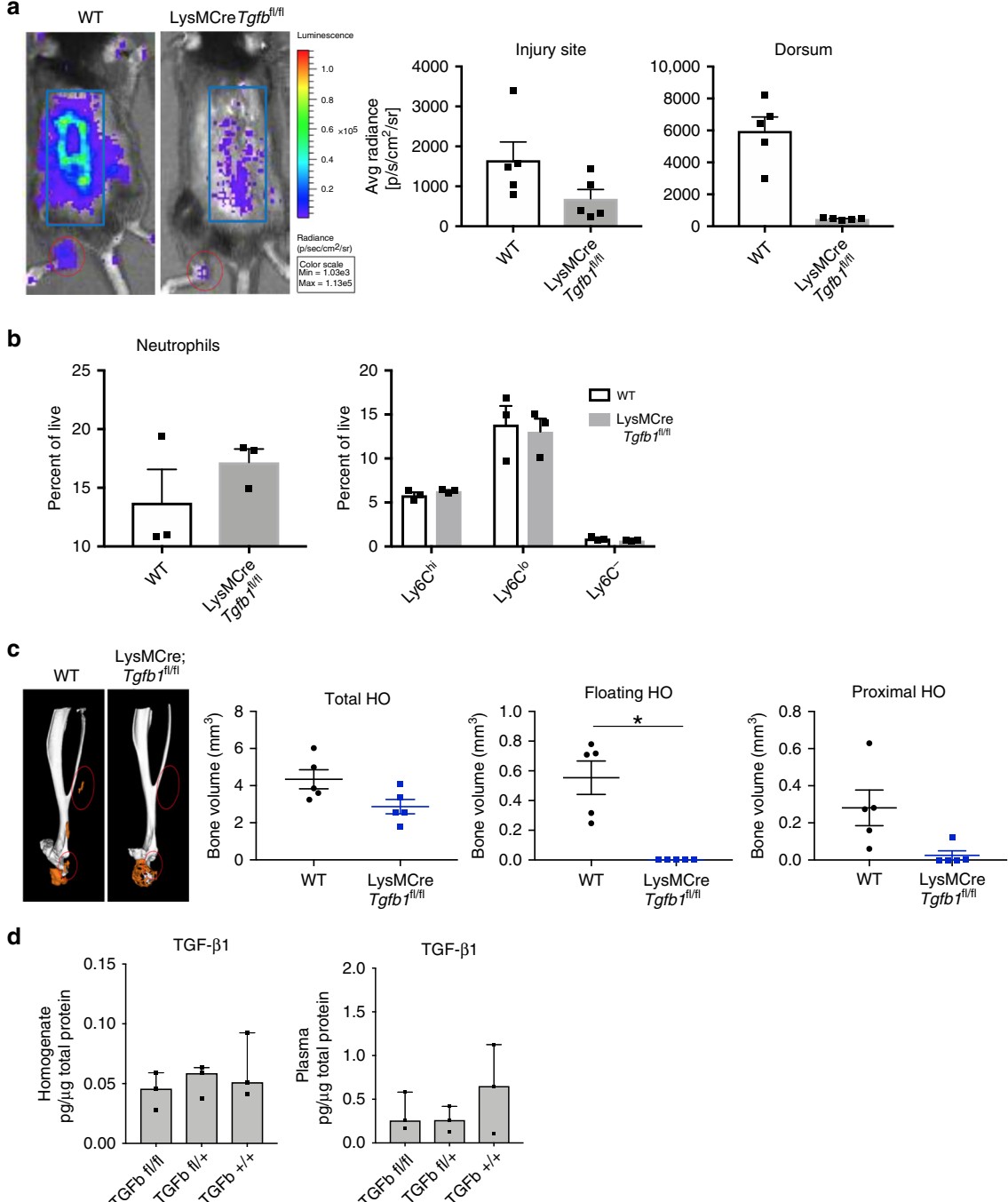

**Fig. 7 TGF-β1 expression in monocyte-derived macrophages contributes to their pathological phenotype and HO formation. a** Representative images of IVIS analysis for MPO of C57BL/6 J and LysMCre-*Tgfb1*fl/fl mice bred on a C57BL/6 J background at 1 week after burn/tenotomy. **Right:** quantification of total bioluminescence/region of interest. $n = 5$/group. Injury site: $t = 1.898$, df = 8, $p = 0.094$; dorsum: $t = 6.385$, df = 4.007, $p = 0.003$. **b** Quantification of neutrophils and monocyte subpopulations based on Ly6C using flow cytometry of injury site 1 week after burn/tenotomy in C57BL/6 J and LysMCre-*Tgfb1*fl/fl mice bred on a C57BL/6 J background. $n = 4$ mice/group. $n = 3$/group. Neutrophils: $t = -1.125$, fd = 4, $p = 0.324$; Ly6C⁻: $t = 1.685$, df = 4, $p = 0.167$; Ly6Clo: $t = 0.315$, df = 4, $p = 0.768$; Ly6Chi: $t = -1.272$, df = 4, $p = 0.272$. **c** MicroCT analysis of tenotomy site 9 weeks after burn/tenotomy in C57BL/6 J and LysMCre-*Tgfb1*fl/fl mice. **Left:** Representative 3D reconstruction. **Right:** Quantification of total HO, floating HO (not associated with tibia or calcaneus) and proximal HO (all HO proximal to calcaneus). $n = 5$/group. Total HO: $t = 2.290$, df = 0, $p = 0.051$; Floating HO: $t = 4.591$, df = 4, $p = 0.008$; Proximal HO: $t = 2.578$, df = 8, $p = 0.033$. **d** Levels of TGFβ1 in pg/ug total protein from homogenates from the extremity injury and plasma from LysMCre-*Tgfb1*fl/fl, LysMCre-*Tgfb1*fl/wt, *or* wild type mice 3 days after burn/tenotomy $n = 3$ mice in each genotype. n = 3/group, df across groups = 2. F statistics: homogenate TGFB1: 0.588, plasma TGFB1: 1.008. All pairwise comparisons were analyzed for homoscedasticity and difference in means via Levene's F-test and two-tailed Student's t-test, respectively. Homoscedastic and heteroscedastic multi-group analyses were performed via ANOVA + post-hoc Dunnett's test and Welch's comparison of means + post-hoc Dunnett's T3 test, respectively. *$p < 0.05$, **$p < 0.01$. Source data are provided as a Source Data file.

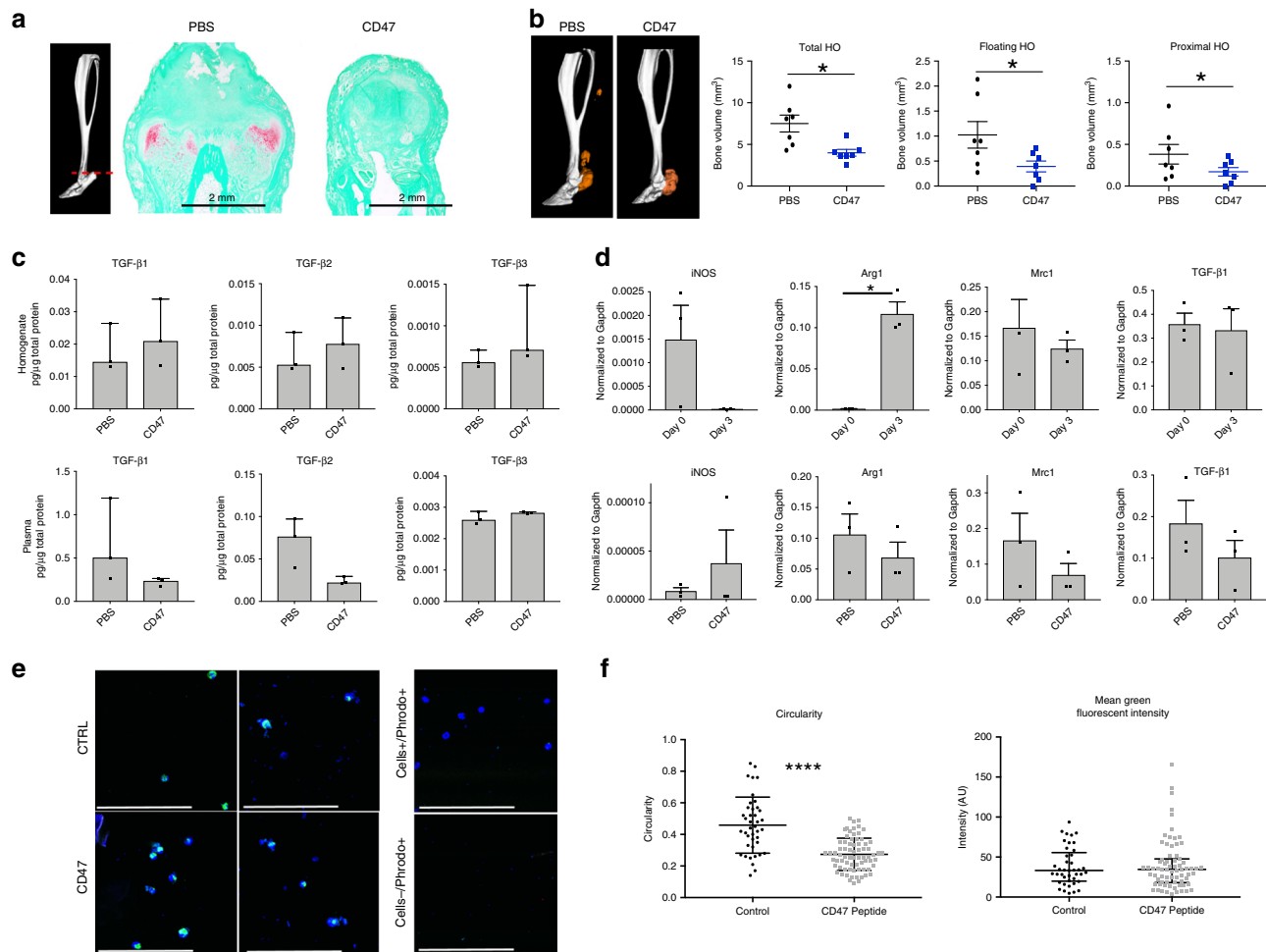

**Fig. 8 CD47-activating peptide treatment alters macrophage phenotype. a** Representative Safranin O stain of tendon injury site 3 weeks after burn/tenotomy in p7N3 (CD47 agonist) treated and PBS control mice. $n = 3$/group. **b** MicroCT analysis of tenotomy site 9 weeks after burn/tenotomy in PBS and p7N3 (CD47 agonist) treated mice. **Left:** Representative 3D reconstruction. **Right:** Quantification of total HO, floating HO and proximal HO. $n = 7$/group. Total HO: $t = 3.415$, df $= 7.840$, $p = 0.009$; Floating HO: $t = 2.201$, df $= 12$, $p = 0.048$; Proximal HO: $t = 2.686$, df $= 8.549$, $p = 0.026$. **c** Levels of TGF-β1, TGFβ2, and TGFβ3 in pg/ug total protein and represented as median with interquartile range from **Top:** homogenates from the extremity injury (TGF-β1: $t = -0.635$, df $= 4$, $p = 0.560$; TGF-β2: $t = -0.643$, df $= 4$, $p = 0.555$; TGF-β3: $t = -1.272$, df $= 2.186$, $p = 0.322$) and **Bottom:** plasma from PBS and p7N3 (CD47) peptide treated mice 3 days after burn/tenotomy (TGF-β1: $t = 1.544$, df $= 2.037$, $p = 0.260$; TGF-β2: $t = 2.747$, df $= 4$, $p = 0.052$; TGF-β3: $t = -1.492$, df $= 4$, $p = 0.210$). $n = 3$ mice per treatment group. **d** qPCR analysis of M1 (*iNos*) and M2 (*Arg1* and *Mrc1*) macrophage markers and *Tgfb1* in macrophages isolated from the extremity injury site of naive (day 0), burn/tenotomy day 3, burn/tenotomy day 3 treated with PBS, and burn/tenotomy day 3 treated with p7N3 (CD47) peptide. Day 0 vs. Day 3—*iNOS*: $t = 2.020$, df $= 2$, $p = 0.181$; *Arg1*: $t = -6.084$, df $= 3$, $p = 0.009$; *Mrc1*: $t = 0.703$, df $= 4$, $p = 0.521$; *Tgfb1*: $t = 0.253$, df $= 4$, $p = 0.812$. PBS vs. CD47 - *iNOS*: $t = -0.834$, df $= 2.043$, $p = 0.491$; *Arg1*: $t = 0.895$, df $= 4$, $p = 0.421$; *Mrc1*: $t = 1.176$, df $= 4$, $p = 0.305$; *Tgfb1*: $t = 1.186$, df $= 4$, $p = 0.301$. **e** Representative images of phagocytosis assay using macrophages isolated from the extremity injury at day 3 after burn/tenotomy in mice treated with PBS or p7N3 (CD47). PBS $n = 3$, CD47 $n = 4$ approximately 25 cells/mouse. **f** Measurement of cellular circularity Circularity: $t = 6.119$, df $= 55.537$, $p = 0.000$ and quantification of mean fluorescent intensity phagocytosed by each macrophage. MFI: $t = -0.357$, df $= 111$, $p = 0.722$. Source data are provided as a Source Data file.

Further, QPCR analysis of markers of M1 (*iNos*) or M2 (*Arginase, Mrc1*, and *Tgfb1*) from sorted macrophages from the injury site of day 0, and 3 days post burn/tenotomy, and day 3 post burn/tenotomy with PBS or p7N3 treatment revealed that with p7N3 treatment, the phenotypic characteristics of the macrophages changed at the injury site (Fig. 8d). As expected, p7N3 treatment resulted in decreased levels of *Tgfb1* expression in macrophages; however, while the cells still expressed *Arg1* and *Mrc1*, the levels were decreased and this was accompanied by increased *iNos* expression, suggesting a change in phenotype of these cells (Fig. 8d). There was no difference in phagocytosis capacity of macrophages sorted from p7N3 treated compared to PBS treated animals (Fig. 8e); however, there was a difference in circularity of these cells (Fig. 8f). Protein analysis of the homogenates revealed

significant increases in Eotaxin and CCL2 in p7N3 treated animals. Interestingly, there was also an increase in LIF, an IL-6 class cytokine that prevents stem cell differentiation, however this did not reach statistical significance (Fig. 9a).

To investigate the effects of p7N3 treatment on the local cellular milieu at the injury site, we performed single cell sequencing on cells isolated from burn/tenotomy mice treated with either p7N3 or PBS. tSNE analysis identified 15 individual clusters in both the PBS and p7N3 treated groups (Fig. 9b, c). Similar to before, granulocytes, monocytes and macrophages comprised the majority of the immune cells present in both groups. To test the effect of systemic p7N3 treatment on monocytes and subsequent macrophages, analysis of the differences in monocyte/macrophage subsets between PBS and

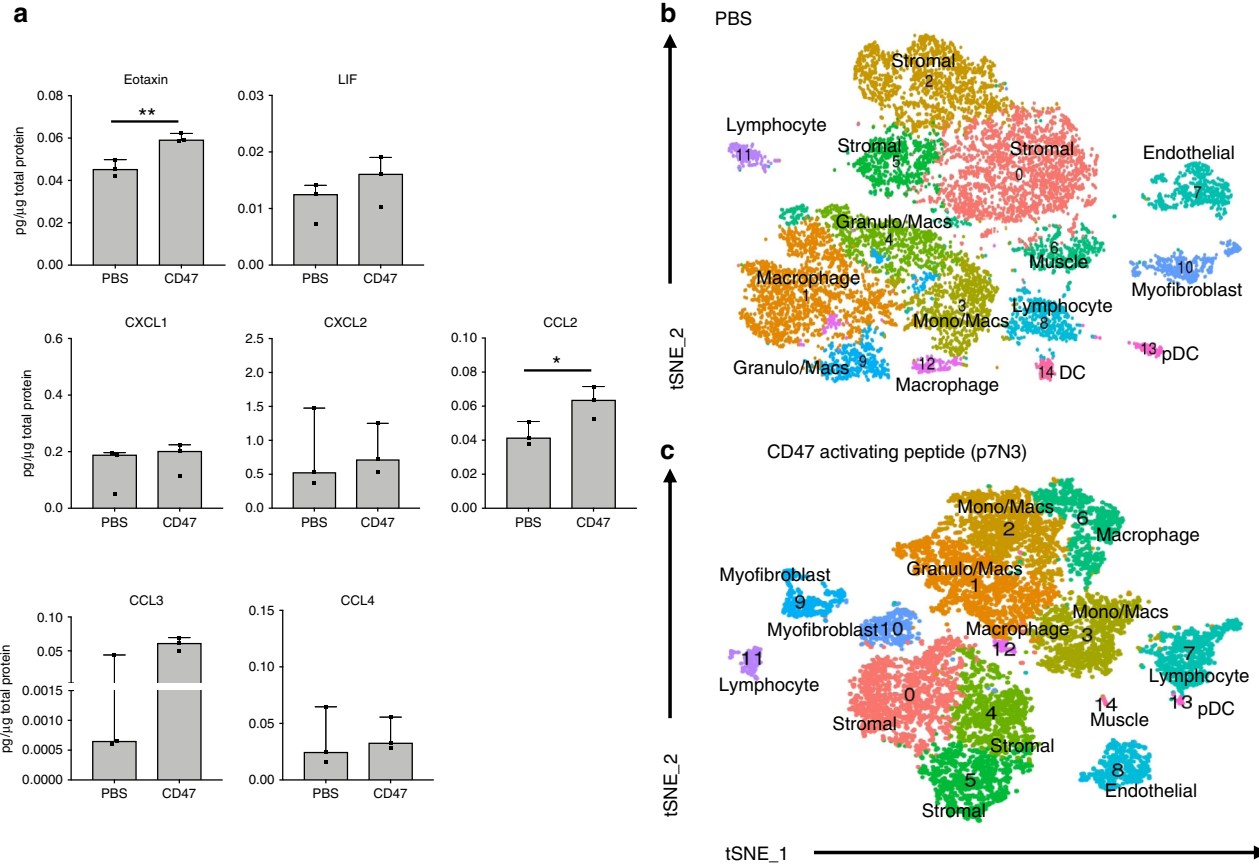

**Fig. 9 CD47-activating peptide treatment alters the transcriptional profile of macrophages. a** Changes in cytokine and chemokine levels from homogenates in mice treated with PBS or p7N3 (CD47-activating peptide) 3 days after burn/tenotomy. Eotaxin: $t = -5.838$, df $= 4$, $p = 0.004$; LIF: $t = -1.162$, df $= 4$, $p = 0.310$; IL15: $t = -2.550$, df $= 4$, $p = 0.063$; CXCL1: $t = -0.604$, df $= 4$, $p = 0.578$; CCL2: $t = -2.870$, df $= 4$, $p = 0.045$; CCL3: $t = -1.492$, df $= 2$, $p$-0.274; CCL4: $t = -0.218$, df $= 4$, $p = 0.838$. **b, c** t-SNE dimensionality reduction analysis of single cell sequencing performed on cells from 1 week burn/tenotomy harvested from the site of extremity injury revealed 15 distinct cell clusters in both **b** PBS and **c** p7N3 (CD47) treated mice (representative performed in triplicate). Source data are provided as a Source Data file.

p7N3 treatment groups was performed. We noted that while there were 5 clusters appearing in PBS and p7N3 treated groups, they were not equivalent. Three of the five clusters had overlapping gene expression profiles (Supplemental Table 2), while two were much different (Supplemental Table 3). Similar to the QPCR analysis above (Fig. 8d), cluster 12 in the p7N3 treated group had high expression of *Nos2*, where there was no *Nos2* expressed in any of the PBS treated monocyte/macrophage clusters. This cluster also had expression of *Cxcl3, Ccl3, Csf3, Il1a*, and *Arg1*. Interestingly, this population had high expression of Activin A (*Inhba*; Supplemental Table 3). Activin A has been shown pathogenic in the genetic form of HO, fibrodysplasia ossificans progressiva, however, M1 macrophages have been shown to secrete large amounts of functional activin A promoting the expression of more M1 markers, while also impairing the acquisition of M2 markers[34]. The second divergent cluster had increased expression of markers *Folr2, Mrc1*, and *Ccl12* (Supplemental Table 3), and appeared similar to the resident macrophage like phenotype we observed in 5a. Interestingly, cluster 12 in the PBS treated group displayed an interferon signature similar to that described by Dick et al. in infarcted tissues from a mouse model of myocardial infarction[22] and by Lin et al. during atherosclerotic progression[20]. Canonical correlation analysis of the two datasets combined demonstrated 16 transcriptionally unique cell clusters (Fig. 10a). We found that stromal cell populations (clusters 1, 3, 5, 6, 7, and 12) from the p7N3 treated group had decreased expression levels in chondrogenic and

osteogenic markers *sox9, runx2, acan*, and *col2a1* in cluster 3 and 4 compared to the PBS control (Fig. 10b). This was consistent with a decrease in Sox9, a chondrogenic marker, signaling at the HO site in CD47 treated mice as shown by immunostaining (Fig. 10c). Together with the results from above, these findings suggest that in the burn/tenotomy model of trauma induced HO, monocyte-derived macrophages modulate the fate of mesenchymal stem cell populations. Systemic treatment with CD47-activating peptide alters the monocyte/macrophage phenotype, decreases chondrogenic and osteogenic markers, and suppresses HO formation (Fig. 10D). CD47 is known to interact with the immunoglobulin superfamily receptor SIRPα[35], in order to determine whether this interaction in macrophages from the HO site in our burn/tenotomy model is important in their M2 polarization, we inhibited SIRPα using a blocking antibody and performed burn/tenotomy. Blocking the CD47/SIRPα interaction during burn/tenotomy injury did not result in increased *Nos2* or decreased *Arg1, Mrc1*, or *Tgfb1* that was seen when mice were treated with the CD47-activating peptide p7N3 (Fig. 10e). This suggests that the interaction of CD47/SIRPα is not important to the M2 polarization of the macrophages during burn/tenotomy, and further, the mechanism of action of the CD47-activating peptide p7N3 is not by disrupting the interaction of CD47 with SIRPα. Therefore, future studies are still needed to further elucidate the mechanism by which activating CD47 on the surface of macrophages can alter their phenotype, and thus impact MSC fate.

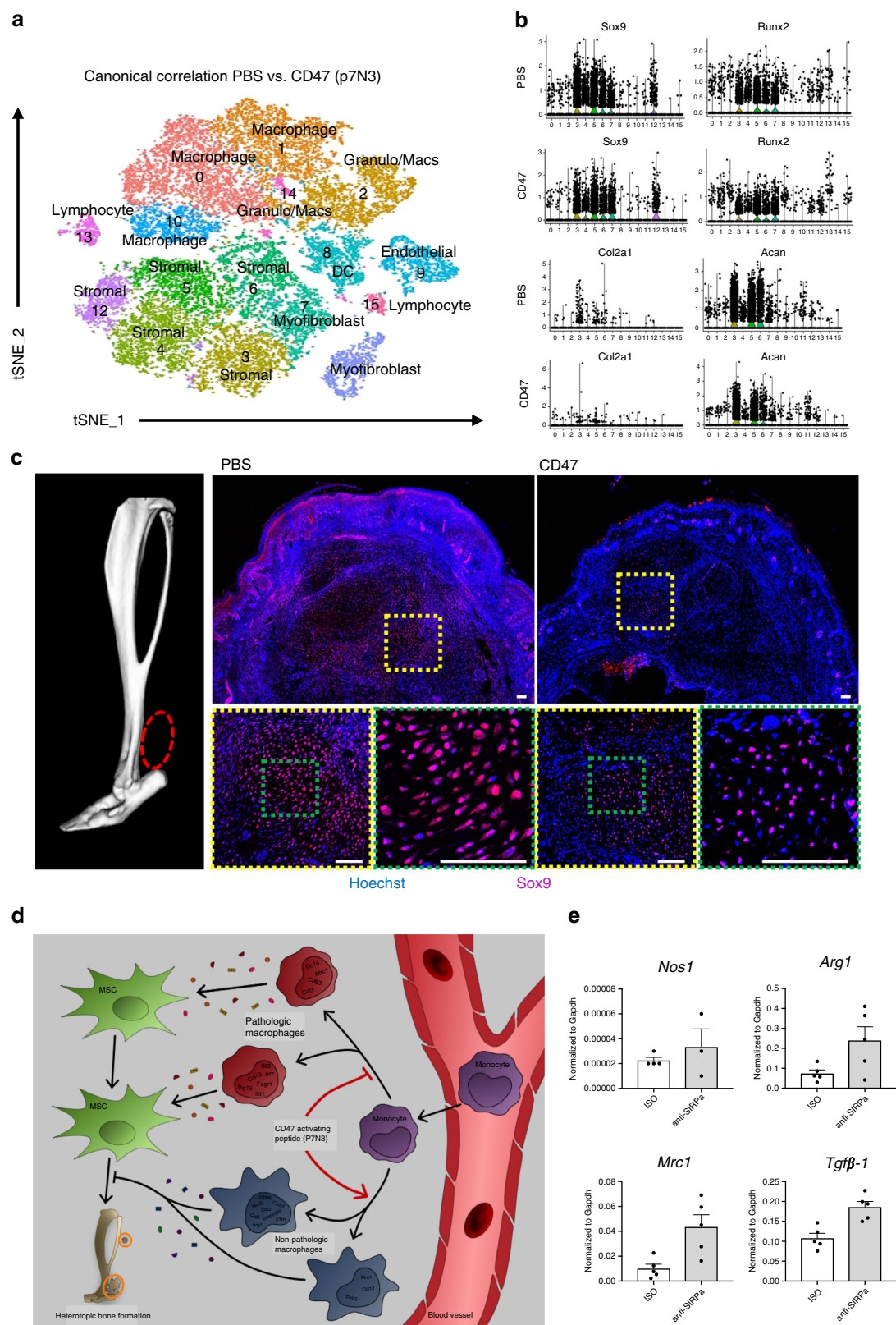

## Discussion

Despite an improved understanding of the progenitor cell niche capable of multi-lineage differentiation after musculoskeletal injury, little has been described about how this response is driven by the interaction with inflammatory cells to facilitate regeneration versus aberrant healing[4]. Recruited circulatory monocytes establish a regenerative milieu at the site of injury through secretion of a myriad of cyto/chemokines. However, monocytes and macrophages are a heterogeneous population of cells comprised of functionally distinct subpopulations and their accurate identification has been hampered by a lack of sensitive biological tools. In this study, we identify cytokines that are expressed in the

**Fig. 10 CD47-activating peptide treatment alters mesenchymal stem cell differentiation independent of the action of SIRPα. a** PBS and p7N3 (CD47 agonist) treatment combined canonical correlation analysis and T-distributed stochastic neighbor embedding (t-SNE) plot identified 16 distinct cell clusters based on gene expression differences. **b** Violin plots for expression of chondrogenic and osteogenic genes. **c** Immunofluorescence for SOX9 in p7N3 (CD47 agonist) treated and PBS vehicle control specimens 10 days after burn/tenotomy. Scale bar represents 100 μm. All pairwise comparisons were analyzed for homoscedasticity and difference in means via Levene's F-test and two-tailed Student's t-test, respectively. *$p < 0.05$, **$p < 0.01$. **d** Cartoon diagram of macrophage participation in the formation of HO and mechanism of action of p7N3, CD47-activating peptide treatment. **e** Normalized expression of Nos2, Arg1, Mrc1, and Tgfβ1 from macrophages isolated from the HO site after burn/tenotomy and treatment with vehical (veh) or anti-SIRPα. Source data are provided as a Source Data file.

initial phase of the inflammatory response and characterize the inflammatory cells recruited in a model of musculoskeletal trauma leading to aberrant regeneration. Formation of HO has been shown to require the recruitment of macrophages and mast cells for genetic and neurogenic HO, however, there is limited granularity on how different macrophage subpopulations may contribute to aberrant mesenchymal cell differentiation[9–11]. Our study utilizes single cell RNA sequencing to delineate monocyte/ macrophage subpopulations that are thought to be functionally distinct based on their transcriptional profiles. We observed temporal changes in these subpopulations over the course of the inflammatory response that indicate a close interaction with mesenchymal progenitor cells leading to aberrant healing. While immune modulating therapies have become a mainstay in cancer therapies, they have failed to translate to diseases of non-oncogenic aberrant cell fate[36–39]. Here, we have validated a therapeutic approach that modulates the macrophage phenotype to prevent aberrant musculoskeletal regeneration in a model of HO. We report comprehensive studies demonstrating the necessity, role, function and phenotype of monocytes/macrophage subpopulations at baseline and after musculoskeletal extremity trauma.

To better characterize the contribution of inflammatory cells during initiation and expansion of HO, we analyzed the injury site at steady state and during a time course following burn/ tenotomy injury that models traumatic HO formation. Neutrophils appeared early after injury but declined during the expansion phase with an influx of monocytes. We then demonstrated that the non-classical Ly6Clo monocyte subpopulation remained present at the injury and appear to play a more central role in aberrant regeneration than the Ly6Chi population.

Though recent studies have implicated macrophages in genetic and neurogenic HO[9–11], depletion studies performed with clodronate do not characterize monocyte subpopulations from which the macrophages derive, nor do they describe the function of the macrophages in the HO microenvironment. In order to obtain an unbiased characterization of the subpopulations of monocytes/macrophages contributing to HO, we performed high throughput RNA sequencing on over 5000 individual cells at multiple key time points after injury. Our study discovered multiple macrophage subpopulations at baseline (no injury), early time points (3 and 7 days after injury) and late time points (21 days after injury) of which some could be attributed to M1 and M2 phenotypes as described by Mantovani et al.[40], however, other subpopulations appeared to not conform and constitute previously uncharacterized subpopulations. Our findings reveal a high level of macrophage heterogeneity suggesting the existence of unexplored populations warranting further functional investigation. Furthermore, single cell transcriptional analysis revealed predetermination of aberrant chondrogenic cell fate of the mesenchymal progenitor cell population as early as 3 days after injury indicating that the inflammatory phase is the critical time window for therapeutic targeting.

In order to devise a macrophage directed therapy that would not abrogate phagocytosis and oxidative burst capacity of

macrophages, which is necessary for normal wound healing, we set to target macrophage phenotype and function. We identified Tgfb1 producing macrophages as a critical link between inflammation and aberrant mesenchymal differentiation, it is expressed by recruited macrophages and its absence in myeloid cells significantly attenuated the aberrant formation of HO after musculoskeletal trauma. Previous reports have suggested that activation of CD47 can reduce the expression of Tgfb1[32,33]. We found that systemic administration of a CD47-activating peptide after injury altered macrophage subset phenotypes. These cells appeared to carry characteristics of both, M1 and M2 phenotypes, according to traditional markers such as Arg1, Nos2, Mrc1, and Tgfb1. This treatment also decreased chondrogenic differentiation of the mesenchymal cell subpopulations and attenuated HO formation. These findings were not due to a simple reduction in Tgfb1 expression, as the levels of TGF-β1 at the injury site and serum were not significantly affected. Rather, it indicates a modulatory effect on the macrophage phenotype leading to a more regenerative healing process. Together, our observations provide a model to understand the interaction between immune cells and local progenitor cells at a site of musculoskeletal extremity injury. Future studies will be needed to further elucidate the role of macrophage phenotype, interactions with MSCs that would drive or prevent osteogenic differentiation, and the formation of HO after severe trauma.

Altogether, our findings reveal a critical role for the interaction of macrophage subpopulations with mesenchymal progenitor cells that governs aberrant stem cell fate. Understanding the myeloid specific signals driving mesenchymal cell fate early in extremity trauma is critical to understanding regenerative biology. While the model in our studies employs a tendon injury, tissues present in other musculoskeletal sites likely have significant overlap in the subpopulations of mesenchymal cells and myeloid cells. Once thought of as reparative, we believe that heterogeneous macrophages are central mediators of progenitor cell fate, and thus could offer novel therapeutic targets for HO or other musculoskeletal injuries in the future. Improved delineation of the functional heterogeneity in recruited inflammatory cells may allow for targeted modulation and control of mesenchymal cell differentiation to restore regenerative pathways and avoid pathologic healing.

## Methods

**Animal use.** All animal procedures were carried out in accordance with the guidelines provided in the Guide for the Use and Care of Laboratory Animals from the Institute for Laboratory Animal Research (ILAR, 2011) and were approved by the Institutional Animal Care and Use Committee (IACUC) of the University of Michigan (PRO0007930). All animals were housed in IACUC-supervised facilities at 18–22 °C, 12-h light–dark cycle with ad libitum access to food and water. For all in vitro and in vivo studies requiring wild-type mice, young adult male (6–10 weeks old) C57BL/6J mice were purchased from Jackson Laboratories (Bar Harbor, ME).

**Transgenic strains.** C57BL/6J background LysM-Cre/Tgfb1fl/fl (compared to strain and age matched control), LysM-Cre/iDTRfl/fl (compared to littermate control) and LysM-Cre/mTmGfl/fl were bred by crossing the respective alleles and then crossing heterozygous mice for both traits to obtain mice undergoing experiments. Littermates were used as controls where possible. Tail specimens were genotyped

using PCR methods and through commercial genotyping resources (Transnetyx, Cordova, TN, USA).

**Burn/tenotomy HO model.** A partial-thickness scald burn injury was administered to animals according to a previously described protocol by a single surgeon.[41,42] Briefly, mice were anesthetized with inhaled isoflurane. Dorsal hair was closely clipped, and an aluminum block heated to 60 °C was exposed to the dorsal region over 30% of the total body surface area for 18 s to achieve a partial-thickness burn injury. Each mouse then received a concurrent sterile dorsal hind limb tendon transection at the midpoint of the Achilles tendon (Achilles tenotomy) with placement of a single 5-0 vicryl suture to close the skin. Pain management was achieved with subcutaneous injections of buprenorphine (Buprenex, Reckitt Benckiser Pharmaceuticals) every 12 h for 2 days. In certain groups, mice were treated with clodronate (200 μl i.v. every 2-3 days), CD47-activating peptide (10 mg/kg i.p. daily), or anti-SIRPα/isotype control (Bioxcell 100 μg i.p. on days −2 and 0).

**Flow cytometry.** After burn/tenotomy the soft tissue around the injury site was dissected from the posterior compartment between the muscular origin and calcaneal insertion of Achilles tendon at the indicated time points. Corresponding soft tissue including intact Achilles' tendon was also harvested from the uninjured contralateral hind limb. Tissue was digested for 45 min in 0.3% Type 1 Collagenase and 0.4% Dispase II (Gibco) in Roswell Park Memorial Institute (RPMI) medium at 37 °C under constant agitation at 120 rpm. Digestions were subsequently quenched with 10% FBS RPMI and filtered through 40μm sterile strainers. Specimens were blocked with anti-mouse CD16/32 and subsequently stained using the following antibodies: FITC:Ly6C, BV510:CD11b, APCH7:Ly6G, BB700:F4/80 (BD), BV421:CD206 and BV650:MHCII (BioLegend). Stained and washed samples were run on a FACSAria II for cell sorting or LSRFortessa for analysis (BD) and analyzed using FlowJo software.

**Immunofluorescence.** Histologic evaluation was performed at indicated time points in wild-type and LysM-Cre/mTmG$^{fl/fl}$ tandem reporter mice following burn/tenotomy. HO anlagen was carefully dissected from injured hind limbs and fixed in 4% paraformaldehyde for 8–12 h in 4 °C. Specimens were cryoprotected with 25% sucrose overnight, embedded in OCT, and cut into 7μm sections. All specimens except for F4/80 and TGFB1 underwent citrate buffered heat antigen retrieval at 70 °C overnight. Sections were subsequently blocked with 2% serum, 1% BSA, and 0.1% fish skin gelatin. Sections were labeled with the following antibodies: rat anti-PDGFRa (santa cruz sc-338, 1:100), rabbit anti-sox9 (abcam ab185230, 1:200), rabbit anti-TGFB1 (Novus NBP1-45891, 1:100), rat anti-CD16/32 (BD 5531410, 1;100), goat anti-CD206 (RD AF2535, 1:50), rat anti-F4/80 (abcam ab6640, 1:50), and rabbit anti-pSMAD3 (Novus NBP1-77836, 1:100), donkey anti-rat alexa fluor 594, 1:500 (Invitrogen 40-0800) and donkey anti-rabbit alexa fluor 488, 1:200 (Invitrogen A-21208). Primary and fluorophore conjugated secondary antibodies were diluted 1:50 (2–4 μg/mL) and 1:200 respectively. Nuclear counterstain was performed with Hoechst 33342 (Life Technologies). Appropriate primary antibody and biological negative controls were run simultaneously with each tested sample.

Antibody labeled sections were imaged with epifluorescent upright scope (Olympus BX-51) equipped with DAPI, single-, and dual-cube 488 nm/TRITC filters attached to an Olympus DP-70 high-resolution digital camera. Each site was imaged in all channels and images were overlaid in Olympus DP Viewer or Photoshop before examination and quantifications in Image J. Images were adjusted only in brightness and contrast identically across comparison groups for clarity where indicated.

**Immunohistochemistry.** 29 cases of heterotopic ossification (HO) were identified in our surgical pathology archives (Johns Hopkins University) under IRB approval, with re-review of all material to verify the diagnostic accuracy (A.W.J.). Three cases with the earliest histologic phase of HO were identified (3/29), including nodular fasciitis-like areas with sheets and fascicles of plump spindled myofibroblastic cells with woven bone formation. 10 μm thick sections were prepared and pretreated with xylene and different concentrations of ethanol. Antigen retrieval was performed by using trypsin enzymatic antigen retrieval solution (Cat no. ab970, Abcam) for 10 mins at room temperature. After washing in PBS, the sections were incubated in 3% hydrogen peroxide for 20 mins for blocking with endogenous peroxidase and followed by washing with PBS. Next, the slides were blocked with 5% goat serum (Cat No. S-1000, Vector Laboratories)) for 1 h and then probed with a rabbit anti-human CD68 (Cat no. ab125212, Abcam) in 1:100 concentration overnight at 4 °C. The next day, the sections were washed off with PBS and the slides were probed with Biotinylated secondary antibody (Cat no. BA-1000, Vector Laboratories, 1:200) and VECTASTAIN Elite ABC HRP reagent (Cat no. PK-6100, Vector Laboratories) for 1 h and 30 min at room temperature. Color development was achieved by treatment with the chromogen DAB (SK-4100, Vector Laboratories) and was carried out for 5-10 min under a microscope. The slides were then rinsed thoroughly in PBS before re-blocking with BIOXALL Blocking Solution (Cat No. SP-6000, Vector Laboratories) for 10 mins and then 2.5% normal house serum for 20 mins at room temperature. After re-blocking the slides, the slides were

incubated with a mouse anti-human TGF-β1 antibody (Cat No. MAB240, R&D Systems) at a dilution of 1:50 overnight. After PBS wash, ImmPRESS AP Anti-Mouse IgG (Cat No. MP-5402, Vector Laboratories) was applied for 30 mins at room temperature and then VECTOR Red Alkaline Phosphatase Substrate Kit (Cat No. SK-5100, Vector Laboratories) was applied for red color visualization. Hematoxyline staining was also done at the end for nuclear staining.

**In vivo bioluminescent imaging of inflammation.** Chemiluminescent activity of myeloperoxidase was measured at the ankle injury site to monitor inflammation. After i.p. injection of the RediJect inflammation probe (150 μl/mouse, 200 mg/kg) animals were imaged using the IVIS Spectrum imaging system (PerkinElmer). Bioluminescence images were acquired for 5 min with F/stop = 1 and binning factor = 8. Planar bioluminescent images were presented in photon/sec/cm$^2$/sr with minimal and maximal thresholds indicated. Living Image software was used to calculate peak total bioluminescent signal at the injury site using through standardized region of interest (ROIs). Data were presented as total flux in photos per second per ROI.

**MicroCT.** Mouse hind limbs were harvested and imaged 9-weeks post-injury (Bruker MicroCT, 35 μm resolution and 357 μA/70 kV/beam). Scans were analyzed by blinded operators manually splining around ectopic tissue and computing volumes at multiple thresholds (GE Healthcare v2.2, Parallax Innovations rc18): unthresholded volume corresponding to total ectopic tissue in addition to 800 hounsfield units (HU) and 1250 HU ectopic bone. In addition to total ectopic bone, analyses were further subdivided between ectopic bone contiguous to or proximal to the tibial-fibular confluence and bone distal to this landmark, defined as proximal and distal HO respectively. Cortical thickness of native bone was measured at the proximal tibia. After identifying the fuse point between tibia and fibula at the mid-diaphysis, a region 0.350–3.800 mm above this anatomical landmark was isolated and then quantified at 1250 HU for mean cortical thickness (Microview 2.2, GE Healthcare Microview).

**Single cell RNA sequencing using 10× genomics.** Tissues harvested from the extremity injury site were digested for 45 min in 0.3% Type 1 Collagenase and 0.4% Dispase II (Gibco) in Roswell Park Memorial Institute (RPMI) medium at 37 °C under constant agitation at 120 rpm. Digestions were subsequently quenched with 10% FBS RPMI and filtered through 40μm sterile strainers. Cells were then washed in PBS with 0.04% BSA, counted and resuspended at a concentration of ~1000 cells/μl. Cell viability was assessed with Trypan blue exclusion on a Countess II (Thermo Fisher Scientific) automated counter and only samples with >85% viability were processed for further sequencing.

Single-cell 3' library generation was performed on the 10× Genomics Chromium Controller following the manufacturers protocol for the v2 reagent kit (10× Genomics, Pleasanton, CA, USA). Cell suspensions were loaded onto a Chromium Single-Cell A chip along with reverse transcription (RT) master mix and single cell 3' gel beads, aiming for 2000–6000 cells per channel. In this experiment, 8700 cells were encapsulated into emulsion droplets at a concentration of 700–1200 cells/ul which targets 5000 single cells with an expected multiplet rate of 3.9%. Following generation of single-cell gel bead-in-emulsions (GEMs), reverse transcription was performed and the resulting Post GEM-RT product was cleaned up using DynaBeads MyOne Silane beads (Thermo Fisher Scientific, Waltham, MA, USA). The cDNA was amplified, SPRIselect (Beckman Coulter, Brea, CA, USA) cleaned and quantified then enzymatically fragmented and size selected using SPRIselect beads to optimize the cDNA amplicon size prior to library construction. An additional round of double-sided SPRI bead cleanup is performed after end repair and A-tailing. Another single-sided cleanup is done after adapter ligation. Indexes were added during PCR amplification and a final double-sided SPRI cleanup was performed. Libraries were quantified by Kapa qPCR for Illumina Adapters (Roche) and size was determined by Agilent tapestation D1000 tapes. Read 1 primer sequence are added to the molecules during GEM incubation. P5, P7 and sample index and read 2 primer sequence are added during library construction via end repair, A-tailing, adaptor ligation and PCR. Libraries were generated with unique sample indices (SI) for each sample. Libraries were sequenced on a HiSeq 4000, (Illumina, San Diego, CA, USA) using a HiSeq 4000 PE Cluster Kit (PN PE-410-1001) with HiSeq 4000 SBS Kit (100 cycles, PN FC-410-1002) reagents, loaded at 200 pM following Illumina's denaturing and dilution recommendations. The run configuration was 26 × 8 × 98 cycles for Read 1, Index and Read 2, respectively. Cell Ranger Single Cell Software Suite 1.3 was used to perform sample de-multiplexing, barcode processing, and single cell gene counting (Alignment, Barcoding and UMI Count) at the University of Michigan Biomedical Core Facilities DNA Sequencing Core.

**Bioinformatics analysis of single cell sequencing data.** A total of ~500 million reads were generated from the 10× Genomics sequencing analysis for each of the replicates. The sequencing data was first pre-processed using the 10× Genomics software Cell Ranger (10× Genomics, Pleasanton, CA, USA); this step includes alignment against mm10 genome. The Cell ranger summary indicated 94% of the input reads as aligned with ~2600 median genes/cell. Further downstream analysis steps were performed using the Seurat R package[43]. We filtered out cells with less

than 500 genes per cell and with more than 25% mitochondrial read content. The downstream analysis steps for each sample type include normalization, identification of highly variable genes across the single cells, scaling based on number of UMI and batch effect, dimensionality reduction (PCA, and t-SNE), un-supervised clustering, and the discovery of differentially expressed cell-type specific markers. Comparison between the sample types was performed by canonical correlation analysis function embedded in Seurat R package. After identifying the common sources of variation between the two datasets, the cells from the two datasets were aligned together and clustered using un-supervised clustering. The differentially expressed genes between the aligned clusters were identified using negative binomial test. For the canonical correlation of PBS and CD47, 12677 and 10,765 cells from the PBS and CD47 groups, respectively, were aligned using un-supervised clustering. Canonical analysis from the time course used a total of 13,364 cells, 3815 from day 0, 3245 from day 3, 3144 from day 7 and 3160 from day 21 that were aligned by un-supervised clustering. Trajectory analysis of the macrophage specific cluster from the canonical analysis of the time course samples (clusters 1, 3, 4, and 7) was performed using Monocle2[44].

**Microarray differential expression analysis**. Severe burn blood microarray data was obtained from GEO (accession: GSE37069). Blood samples post burn ($n = 553$) were compared to healthy control blood samples ($n = 37$) using Limma (citation: https://link.springer.com/chapter/10.1007/0-387-29362-0_23).

**Gene signature GSEA analysis**. For each gene a rank list was generated by ordering each gene in the differential expression analysis by a metric multiplying the limma (Anders and Huber, 2010) log-fold-change value (logFC) by the *p*-value (p.adj). These rank lists were used in a weighted, pre-ranked GSEA (Subramanian et al., 2005) analysis against MSigDBv5 (Liberzon et al., 2011). Significant associations were determined for any gene set having an FWER p-value below 0.01.

**Multiplex Cytokine Analysis**. Cytokine and chemokine levels, including TGFβ1, TGFβ2, and TGFβ3, in plasma and tissue homogenates were measured using luminex multiplex bead-based analysis (EMD Millipore, Burlington, MA). Data were collected on a Bio-Plex 200 system following the manufacturer's protocols. All data shown fell within the linear portion of the appropriate standard curves. Total protein was determined via Bradford assay (Thermo Fisher Scientific) and used to normalize analyte concentrations to pg/mg of total protein.

**RNA isolation and qPCR**. Harvested cells were added to TRIzol reagent (Ambion/Life Technologies) and RNA was purified using RNeasy Mini Kit from (Qiagen) following the manufactures procedure with added DNAse digestion. RNA was converted to cDNA using the High Capacity cDNA Reverse Transcription Kit (Applied Biosystems) following the manufactures procedure. The cDNA was then diluted 1:5. TaqMan FAM-MGB primer/probe mixes (Supplemental Table 4) were obtained from ThermoFischer Scientific. Assays for the housekeeping gene Gapdh (Mm99999915_g1), Arg1 (Mm00475988_m1), Mrc1 (Mm01329362_m1), Nos1 (Mm01208059_m1), and Tgfb1 (Mm01178820_m1) were used. RT-qPCR reaction mix was prepared according to manufactures protocol for TaqMan (Applied Biosystems). In 96-well plates, 5 μl of diluted cDNA and 5 μl of TaqMan master mix was added in a 1:1 ratio for total volume of 10 μLs. Plates were organized to have technical triplicate for each gene expression. The plates were run in an Applied Biosystems 7500 Real Time qPCR cycler. Gene expression was normalized to Gapdh (dCt), and displayed as 2^-(dCt).

**Western blot analysis**. Tissue was homogenized with mortar and pestle. Cells lysed with RIPA buffer lysis system (sc-24948A) and assessed for total protein using Pierce BCA Protein Assay Kit (cat no. 23225). Proteins from lysate were separated on NuPAGE 4–12% bis-tris Gel (NP0335BOX) with NuPAGE SDS running buffer (NP0001). Protein was transferred to PVDF membrane (IPVH00010) in NuPAGE transfer buffer (NP0006). Membrane was washed in wash buffer (1× TBS 0.05% Tween-20). One hour at room temperature in 5% milk protein in wash buffer (block solution) for 1 h. Incubated with primary antibody diluted in 5% BSA in wash buffer. Appropriate HRP linked secondary was diluted in block solution. Original uncropped immunoblots can be found in the corresponding source data file to this manuscript. FIJI and excel were used for quantification.

**Phagocytosis assay**. 50,000 FACS sorted macrophages from 3 day burn/tenotomy mice treated with either PBS or p7N3 were plated in an 8 well chamber slide. These were allowed to settle down and attach to the plate. Cells were treated with pHrodo green *E. coli* bioparticles (Invitrogen) according to the manufacturer's protocol. After 90 min of incubation, cells were washed, and chambers removed. Macrophages Slides were placed into 1× PBS in plastic petri dish and imaged with 25× dipping lens on Leica SP5 Confocal Laser Scanning Microscope (University of Michigan Molecular Imaging Lab). Cells were excited with 405 nm Diode light (5% power) for Hoescht stain and pseudo-colored blue, and at 495 nm (white light laser, 70% power) for pHrodo green STP ester dye. Emitted light was captured by detectors at 415–470 nm and 505–570 nm, respectively. Images were taken of random locations of the slide until about 25 cells per mouse were imaged. Images were then quantified in Image J for percentage of pHrodo+ cells. Images were then separated by blue and green channels and converted into 8-bit images. Full panel auto-thresholding was performed to determine most accurate threshold algorithm. Max entropy algorithm was selected and utilized to auto-threshold images in each channel. Blue and green channel overlays were used to determine cell circularity, roundness, and solidity. Green channel mean florescent intensity was measured to determine pHrodo uptake.

**Statistical analysis**. Primary outcome of interest for a priori power analysis is volume of mature ectopic bone formation. To detect 50% decrease from 7.5 mm³ (with standard deviation of 2.2 mm³) in untreated mice at significance level of 0.05 and power of 0.80, at least three mice were required per group. Means and SD were calculated from numerical data, as presented in text, figures, and figure legends. Bar graphs represent means with error bars specifying one SD. Pairwise comparisons were conducted with Student's *t*-test after confirmation for homoscedasticity via F-Test. *P*-values included in figure legends.

**Reporting summary**. Further information on research design is available in the Nature Research Reporting Summary linked to this article.

## Data availability
Single cell data is deposited in GEO (GSE126060). RNA sequencing data of human data was obtained from GEO and is available online (accession: GSE37069). Mouse RNA sequencing is deposited in GEO (GSE126118). The source data underlying Figs. 1a–e, 2b, d, 3b–d, 3f, 5b, 5e–h, 6b–d, and 6f–g, Supplemental Figs. 1–7, and Supplemental Tables 1–4 are provided as a Source Data file.

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

## Acknowledgements

We would like to thank the University of Michigan Center for Molecular Imaging and Amanda Fair for her assistance. We would further like to thank the Biomedical Research Core Facilities DNA Sequencing Core and the staff for processing sequencing samples. We would also like to thank David Cholok MD, Serra Ucer PhD, Amrita Joshi MS, Talis Rehse BS, Prasanth Kotha, Sidra Kader BS, Mohamed A. Garada BS, and Ramkumar T. Annamalai PhD for their technical assistance. MS was funded by Plastic Surgery Foundation National Endowment Award, AWJ was funded by NIH/NIAMS (R01 AR070773, K08 AR068316), NIH/NIDCR (R21 DE027922), USAMRAA through the Peer Reviewed Medical Research Program (W81XWH-18-1-0121, W81XWH-18-1-0336), Department of Defense through the Broad Agency Announcement (BA160256), American Cancer Society (Research Scholar Grant, RSG-18-027-01-CSM), the Orthopaedic Research and Education Foundation with funding provided by the Musculoskeletal Transplant Foundation, the Maryland Stem Cell Research Foundation, and the Musculoskeletal Transplant Foundation. YM was funded by NIH R01DE020843, DoD W81XWH-11-2-0073; LKM was funded by DK053904. CH was supported by Howard Hughes Medical Institute Medical Research Fellowship. BL was funded by NIH, NIAMS, NIH1R01AR071379, ACS Clowes Award, International Fibrodysplasia Ossificans Progressiva Association Research Award, NIH, NIGMS, K08GM109105, R01GM123069.

## Author contributions

All authors read and reviewed this manuscript. B.L.,: conceived the study, obtained the funding and wrote the manuscript with the help of M.S. and A.K.H. and with input from others below. M.S. and A.K.H: planned and executed experiments, wrote the manuscript, managed the project C.H. Immunofluorescent staining and imaging. Contributed to experimental concepts, assisted in running flow cytometry staining and capture, and assisted in statistical analysis of data. C.P.: MicroCT analysis, cellular imaging and statistical analysis of phagocytosis experiment. Assisted on statistical analysis of data. R.M.: bioinformatics analysis and experimental input on single cell sequencing and trajectory analysis. K.V. S.L, J.L., and N.D.V.: helped execute laboratory experiments, cell culture work, perform surgeries, prepare RNA and prepare samples for sectioning for histological staining. N.P.: cell culture work, helped with RNA and QPCR analysis. S.L. and S.A.: design of in vivo experiments, performing flow cytometry experiments and analysis of flow cytometry data. A.W.J. and J.X.: Performed immunohistochemistry in human samples of heterotopic ossification. L.K.M.: provided LysmCre/Tgfb$^{fl/fl}$ mice and performed experiments for model characterization. Y.N.: RNAseq of human and mouse HO tissue. W.C., D.N., and K.G.: cell culture work and phagocytosis assay. S.K. and Y.M.: project guidance, intellectual input, and thoroughly read and edited the manuscript.

## Competing interests

The authors declare no competing interests.
