## [Peer Review File · Nature Communications]

Reviewers' comments:

Reviewer #1 (Bone immunity, cytokine signaling.)(Remarks to the Author):

This study presents a thorough characterization of the site of heterotopic ossification in a burn tenotomy model, with a focus on immune cells and inflammatory molecules and how they might affect stromal progenitors, using a combination of single cell RNAseq, analysis of protein homogenates, immunofluorescence, in vivo imaging, and flow cytometry (Figs 1-4, which also show that monocyte/macrophage depletion using clodronate significantly decreases HO). The authors then attempt to build on their findings to implicate myeloid-derived TGFb in HO, and to suggest that myeloid cell targeting by a CD47-activating peptide suppresses HO. The first 4 figures are well done, interesting, and advance the field; this represents to my knowledge the highest resolution analysis of the inflammatory response in a musculoskeletal injury model, and thus of general interest. Figures 5 and 6 have very significant issues as the interpretations and conclusions are not well supported by the evidence.

Major points

1. Figure 5. Although TGFb signaling is clearly detected at the injury site and myeloid deletion of Tgfb1 strongly affects the in vivo MPO signal, the effects on cell infiltration and on total HO itself are fairly minimal and not significant (and effects on HO much less than effects of myeloid cell deletion shown in Fig 3). Total TGFb in tissue is not affected (Fig 5h) and the effects of myeloid Tgfb1 deletion on gene expression are not shown. My inclination would be to call this a negative result, and certainly not to state, including in the abstract, that TGFb "was critical for HO".

2. The CD47 activating peptide actually seems to work but does not have a significant effect on TGFb in the tissues. A brief review of public databases reveals that CD47 is expressed in multiple tissues and cell types. Thus, it is equally inappropriate to conclude that CD47 activating peptide is working via effects on myeloid cells, and that it works to "reverse TGF-b1 driven contribution to....HO". The effects of the peptide on myeloid cell populations are interesting (Fig 6h and 6i) but could well be indirect. My inclination would be to state that the peptide affects the inflammatory response and suppresses HO by unclear mechanisms (and that the causal relationship between the changes in myeloid cells and HO phenotype is not known and would need to be experimentally addressed).

Lesser points

3. The authors show some contralateral controls but I could not readily find the sham surgery and partial tenotomy controls mentioned in methods – these controls and results should at least be discussed and preferably shown in supplementary data.

Minor points

4. It is not clear how "chemokines increase other pro inflammatory molecules..." (text, pg. 5), text should be changed to reflect the data.

5. It is unusual that monocyte/macrophage-specific Csfr1 is expressed in cells identified as lymphocytes and granulocytes (Fig 1 f and 1h), this should be commented on.

6. Lysozyme M (LysM) (pg. 7) is not a transcription factor, it is an enzyme.

7. Although clodronate preferentially affects monocytes it can potentially delete phagocytic tissue

macrophages, thus the text on pg. 8 should be modified.

8. The statements about monocytes and macrophages on top of page 9 should be amended (see recent review by David Hume for a perspective on this and potential lineage relationships amongst these cell types)

9. "flox" is usually abbreviated as "fl", not as "fx" as done throughout the manuscript.

10. It is probably worth mentioning that the PBS group in Table 2 contains interferon-stimulated genes.

Reviewer #2 (Macrophage biology, innate programming.)(Remarks to the Author):

General Comment: In this manuscript, Sorkin et al. demonstrate the heterogeneity of monocyte/macrophage populations during post-traumatic responses. The author claim that TGF- β produced by monocytes/macrophages contributes to HO development. Hence, administration of p7N3, a CD47 activating peptide that down-regulates TGF- β expression, can alleviate HO and alter macrophage phenotype. The concept of this study is intriguing, but there remain several concerns which would require substantiations, and additional experiments in order to support their conclusion.

1. As shown in Fig. 1f and 1g, five clusters of monocytes/macrophages have been identified at injury site post B/T. Which one(s) is the primary population(s) producing TGF- β ? Does it resemble a known monocyte/macrophage subset or a novel subset? A better characterization of the monocyte/macrophage subset would improve the significance of this work.

2. Fig. 1e indicates elevated TGF- β 2 expression after B/T. However, based on the data shown in Fig. 1f, TGF- β 2 expression cannot be found in any cell clusters. Please explain this discrepancy.

3. CSF1R (M-CSFR) should be expressed by the cells of monocyte lineage. Why do all leukocyte clusters (Fig. 1h), even including neutrophils and lymphocytes, exhibit marked expression of Csf1r?

4. The gating strategy for FACS analysis in Fig 2d seems arbitrary and inconsistent, because the size and the position of the gates are varying from sample to sample. Furthermore, the authors claim that "Ly6Clo cells also co-expressed the markers F4/80 and CD206 (page 8)". Thus, co-staining of F4/80 and Ly6C as well as co-staining of CD206 and Ly6C should be displayed.

5. In addition to reduced MPO activity and ankle size shown in Fig. 3, the authors need to test whether the production of representative cytokines and chemokines that participate in HO is affected by monocyte/macrophage depletion. Especially, it should be determined whether the level of TGF- β is altered.

6. The involvement of TGF- β in HO has been documented in the literature, so one of the aims and novelties of the present study is to identify the cellular origin of TGF- β . However, the data shown in Fig. 5 do not fully address this question, and the experiment is not well designed and interpreted.

First, it should be shown that TGF- β production is successfully depleted in the monocytes/macrophages of LysMCre-Tgfbfx/fx mice, because the TGF- β levels of these mice are comparable to WT counterparts (Fig. 5h).

Second, LysMCre mice are used for gene depletion in myeloid cells, including monocytes, macrophages and granulocytes. Neutrophils also produce considerable amount of TGF- β . Therefore, LysMCre-Tgfbfx/fx mouse, in which TGF- β expression is abolished in neutrophils as well, is not an ideal model to study the contribution of monocytes/macrophages derived TGF- β . Because TGF- β is mainly produced in F4/80+ macrophages (Fig. 5c), the authors may use F4/80Cre mice to perform the experiment.

7 Some panels in Fig. 1 are mislabeled. Fig. 1b and Fig. 1c should be switched; Fig. 1d and Fig. 1e should be switched.

Reviewer #1 (Bone immunity, cytokine signaling.):

Major points

1. Figure 5. Although TGFb signaling is clearly detected at the injury site and myeloid deletion of *Tgfb1* strongly affects the in vivo MPO signal, the effects on cell infiltration and on total HO itself are fairly minimal and not significant (and effects on HO much less than effects of myeloid cell deletion shown in Fig 3). Total TGFb in tissue is not affected (Fig 5h) and the effects of myeloid *Tgfb1* deletion on gene expression are not shown. My inclination would be to call this a negative result, and certainly not to state, including in the abstract, that TGFb “was critical for HO”.

We thank the referee for this comment. We agree that there was no difference in the protein levels of TGFb in the homogenates of the injury site, however there was a decrease, albeit not significant, in the levels in the plasma suggesting that the circulating cell production may be decreased. In our BT model we have shown that there are 2 different types of HO formed. Proximally, we observe non-bone associated or “floating” HO which derives from the cut end of the Achilles tendon. We believe it is significant that *Tgfb1* knockout in the LysMCre lineage completely mitigated this floating HO. Future studies will further characterize and explore the heterogeneity in HO progenitor cells, but we think it is beyond the scope of this study. We understand the reviewer’s reservation in the wording we used and have changed some of the wording. We believe TGFb producing macrophages are important in the formation of HO after traumatic injury, and by altering the expression of TGFb from these cells we effect their phenotype that results in a different cell functionally.

2. The CD47 activating peptide actually seems to work but does not have a significant effect on TGFb in the tissues. A brief review of public databases reveals that CD47 is expressed in multiple tissues and cell types. Thus, it is equally inappropriate to conclude that CD47 activating peptide is working via effects on myeloid cells, and that it works to “reverse TGF-b1 driven contribution to....HO”. The effects of the peptide on myeloid cell populations are interesting (Fig 6h and 6i) but could well be indirect. My inclination would be to state that the peptide affects the inflammatory response and suppresses HO by unclear mechanisms (and that the causal relationship between the changes in myeloid cells and HO phenotype is not known and would need to be experimentally addressed).

The points addressed by this comment is well taken, we have edited the manuscript upon the suggestion of the reviewer (Page 2, 15, 16, and 19)

Lesser points

3. The authors show some contralateral controls, but I could not readily find the sham surgery and partial tenotomy controls mentioned in methods – these controls and results should at least be discussed and preferably shown in supplementary data.

We apologize but this was a typo in our methods section and the edits have been made the text removed.

Minor points

4. It is not clear how “chemokines increase other pro inflammatory molecules...” (text, pg. 5), text should be changed to reflect the data.

It has been shown previous that MIP1a (Ccl3) and MIP1b (Ccl4) are able to induce the synthesis of proinflammatory cytokines IL-1, IL-6, and TNF- α (Fahey TJ et al. 1992). However, conversely TNF- α and IL-1 β can also induce the expression of Ccl3 (Lukacs N et al. 1995, McManus C 1998 et al. and Miyamoto Y et al. 1999). We have changed the wording in this section and added references (page 5) to support these statements.

5. It is unusual that monocyte/macrophage-specific *Csfr1* is expressed in cells identified as lymphocytes and granulocytes (Fig 1 f and 1h), this should be commented on.

We thank the reviewer for this thought-provoking subject. A study using EGFP expression driven by the *csfr1* promoter demonstrated that transcripts of EGFP and *Csfr1* could be found on both macrophages and granulocytes, however, while transcripts were seen in granulocytes, protein product was not (Hume D. et al 2007). Additionally, it was shown that the EGFP expression was upregulated in all dendritic cell (DC) subsets during differentiation (MacDonald et al. 2005). This was concluded in the paper using Progenipoinetin treatment of MacGreen mice, an agonist for the G-CSF receptor. Interestingly, the tissue specific levels of G-CSF are increased in our model, therefore, it wouldn't be inconceivable that DCs in the injury site stimulated with G-CSF might produce mRNA for *Csfr1*. Indeed, many of the cells that fell into our lymphocyte clusters were classical DC and plasmacytoid DC in nature. We have added a brief statement as well as cited these papers in order to address both of these positive clusters in our feature plots. In addition, it has been seen that the CSF-1R-driven Cre mouse as described by Deng et al. also deletes in neutrophils and splenic T-cell (Deng et al., 2010). We have addressed this and added a statement about this in our results section (page 6 and page 7).

6. Lysozyme M (LysM) (pg. 7) is not a transcription factor, it is an enzyme.

We apologize for this error; we have now corrected this in the manuscript (page 7).

7. Although clodronate preferentially affects monocytes it can potentially delete phagocytic tissue macrophages, thus the text on pg. 8 should be modified.

We have added a statement and a reference to address the reviewer's statement (page 9).

8. The statements about monocytes and macrophages on top of page 9 should be amended (see recent review by David Hume for a perspective on this and potential lineage relationships amongst these cell types)

We thank the reviewer for bringing this new review to our attention. We have edited the paper removing the statement "...that largely derive from Ly6C^{lo} monocytes", as we do not know for certain whether macrophages came from directly from these cells.

9. "flox" is usually abbreviated as "fl", not as "fx" as done throughout the manuscript.

We have changed all the fx to fl in our manuscript (pages 7, 12)

10. It is probably worth mentioning that the PBS group in Table 2 contains interferon-stimulated genes.

We thank the reviewer and also agree that there is indeed a population in the PBS treated group that has an interferon signature. This was initially interesting to us, as a similar population was recently reported in a model of myocardial infarction injury in Nature Immunology (Dick et al. 2019 Nat Imm). We added a statement about this cluster in our results section (page 16).

Reviewer #2 (Macrophage biology, innate programming):

1. As shown in Fig. 1f and 1g, five clusters of monocytes/macrophages have been identified at injury site post B/T. Which one(s) is the primary population(s) producing TGF- β ? Does it resemble a known monocyte/macrophage subset or a novel subset? A better characterization of the monocyte/macrophage subset would improve the significance of this work.

We thank the reviewer for this valuable comment. "Clusters 0, 1, and 3 represent the clusters with most of the TGF- β 1 gene expression. Analyzing the top genes specific in these clusters and doing a literature search of other single cell analysis, we find that populations similar to those seen in our clusters 1 and 3 have been described. Gene signature similar to our cluster 1 has been seen in atherosclerosis and acute lung injury, while signatures similar to cluster 3 were seen in myocardial infarction and atherosclerosis. Cluster 0 does appear a bit more unique. This cluster appears similar to that seen when analyzing a circulating monocyte. This might signify an early infiltrating monocyte, or perhaps have a unique signature as a result of the peripheral response to the burn before entering the tenotomy site." This more in-depth characterization is now

discussed in the results section, page 6. A data table with the 10 highest expressed genes from each of these clusters has been added as Supplementary Table 1.

2. Fig. 1e indicates elevated TGF- β 2 expression after B/T. However, based on the data shown in Fig. 1f, TGF- β 2 expression cannot be found in any cell clusters. Please explain this discrepancy.

We appreciate the reviewer's comment. Single cell RNAseq determines transcript level expression of genes. It is possible that protein had already been made and secreted by day 3 when the cells were prepared, and the transcriptional programming for TGF- β 2 had been downregulated in the cells by this time of sequencing. There is a small expression in the fibroblast/myofibroblast population. It is also possible because RNASeq is a relative abundance measurement, and cDNA fragments are sequenced in a random sample fashion, that if a cell population composed of small numbers of cells contributing a small number of random fragments is where TGF- β 2 is expressed that this will be missed in our analysis. We sequenced ~500 million reads from each of our replicates performed, however cells that had less than 500 genes per cell sequenced were excluded from our analysis.

3. CSF1R (M-CSFR) should be expressed by the cells of monocyte lineage. Why do all leukocyte clusters (Fig. 1h), even including neutrophils and lymphocytes, exhibit marked expression of *Csf1r*?

Please see the answer to bullet point 5 above raised by reviewer 1. Single cell RNAseq, detects levels of RNA and not necessarily protein. It has been described the expression of *Csfr1* RNA in cell subsets that fall within these clusters, this could account for the positive expression we are seeing. Statements to clarify this, along with references have been made in the manuscript, page 6 and 7.

4. The gating strategy for FACS analysis in Fig 2d seems arbitrary and inconsistent, because the size and the position of the gates are varying from sample to sample. Furthermore, the authors claim that "Ly6C^{lo} cells also co-expressed the markers F4/80 and CD206 (page 8)". Thus, co-staining of F4/80 and Ly6C as well as co-staining of CD206 and Ly6C should be displayed.

We understand the reviewer's concern. This was a time course experiment of our B/T model and the data for each time-point was obtained on different days. The size and position of the gates was determined based on the isotype and single-color controls for these particular days. We have included the negative cell populations gates in Supplemental Figure 2. We have also included the requested plots to Figure 2d showing the expression of F4/80 and showing the co-expression of CD206⁺ and F4/80 with the Ly6C^{lo} population.

5. In addition to reduced MPO activity and ankle size shown in Fig. 3, the authors need to test whether the production of representative cytokines and chemokines that

participate in HO is affected by monocyte/macrophage depletion. Especially, it should be determined whether the level of TGF- β is altered.

The reviewer's comment is well taken. We have analyzed the injury site homogenate and plasma at day 3 from LysMCre-iDTR and litter mate control mice, where monocytes/macrophages were depleted by pre-injection of diphtheria toxin (DT) two days before the B/T, the day of B/T and at day 2 after the B/T. We find that the major changes with regards to pro-inflammatory cyto/chemokines are in the serum at day 3. This suggests that the depletion of macrophages effects the systemic response to the dorsal burn but doesn't change the early response occurring at the HO site. We find a trend for decreases in plasma CCL2, G-CSF, GM-CSF, IL1b, IL-6, and TNF-a levels when macrophages are depleted, although nothing reached significance. There was also a decrease in the level of TGF- β 1 at the tenotomy site, albeit not significant (Supplementary Figure 5). This has been added to page 9.

Analysis of the plasma and tenotomy site in LysMCre-Tgfb1 mice was also performed. There were no significant changes in the levels of pro-inflammatory cyto/chemokines at the tenotomy site. There was a trend for decreases in plasma GM-CSF, IL1b, IL-6, CCL3, CCL4, CCL5, but no differences in the levels of TGF- β 1. This suggests that depletion of monocytes or TGF- β 1 producing monocytes alter the peripheral response and subsequent downstream immune cell response which occurs at the injury site. These data points to TGF- β 1 production in the monocyte as more of a marker of the monocyte/macrophages important in potentiating bone formation, and less of the cause. This has been added to Supplementary Figure 6. A section has been added with these results (Page 14).

6. The involvement of TGF- β in HO has been documented in the literature, so one of the aims and novelties of the present study is to identify the cellular origin of TGF- β . However, the data shown in Fig. 5 do not fully address this question, and the experiment is not well designed and interpreted. First, it should be shown that TGF- β production is successfully depleted in the monocytes/macrophages of LysMCre-Tgfbx/tx mice, because the TGF- β levels of these mice are comparable to WT counterparts (Fig. 5h).

We thank the referee for this comment. While the TGF- β levels at the injury site were similar, it should be noted there are other sources of *Tgfb1* besides myeloid cells. Plasma levels in these mice were decreased, suggesting that the circulating monocyte and myeloid cell populations were affected. To address the successful depletion of *Tgfb1* in our mice, bone marrow was flushed from 4-week-old WT (C57B6) or KO (LysMCre-*Tgfb1*^{fl/fl}) mice and macrophages induced in culture with 30ng/ml M-CSF for 5 days. Western blot was performed and revealed a significant decrease in TGF- β 1. This has been added to Supplementary Figure 3. And a statement was added to page 13.

7. Second, LysMCre mice are used for gene depletion in myeloid cells, including monocytes, macrophages and granulocytes. Neutrophils also produce considerable amount of TGF- β . Therefore, LysMCre-Tgfbx/tx mouse, in which TGF- β expression is

abolished in neutrophils as well, is not an ideal model to study the contribution of monocytes/macrophages derived TGF- β . Because TGF- β is mainly produced in F4/80+ macrophages (Fig. 5c), the authors may use F4/80Cre mice to perform the experiment.

We appreciate the suggestion from the reviewer. We have done extensive research on myeloid cell specific cre mouse lines. Vi et al. has shown that LysM-cre effectively targeted macrophage lineage cells during the fracture repair process in vivo (Vi et al. 2015 J.Bone Miner.) Additionally, there was an excellent analysis of multiple myeloid “specific” cre lines published by Abram et al. in 2014. She crossed each of the lines to a ROSA-YFP reporter mouse and demonstrated across a variety of non-stimulated hematopoietic cells, focusing on the myeloid cells, among multiple tissues the efficiency and specificity of each strain. In the blood monocytes, she showed that the LysMcre line had the highest amount of recombination signified by the expression of YFP. F4/80cre is limited only to the F4/80^{hi} macrophages, and in this paper had very little if any recombination.

8. Some panels in Fig. 1 are mislabeled. Fig. 1b and Fig. 1c should be switched; Fig. 1d and Fig. 1e should be switched.

We thank the reviewer for catching this error, we have fixed the labels on the figures.

We would like to thank all reviewers for their detailed and timely review and overall positive assessment of the work. We believe we have addressed all reviewer concerns greatly enhancing the quality of the manuscript and making it of specific interest to the *Nature Communications* readership.

REVIEWERS' COMMENTS:

Reviewer #1 (Remarks to the Author):

The authors have done a thorough job of revision and the manuscript is substantially improved and conclusions better aligned with data.

I still think the conclusions about TGFbeta are overstated, but the authors have toned this down sufficiently that I think it is reasonable for publication.

I agree with the new statement about clodronate liposomes not crossing capillary walls under physiological conditions, what about after injury and vascular leak, and could this affect the results and interpretations?

Overall the manuscript is quite interesting and appropriate for publication in Nature Communications.

Reviewer #2 (Remarks to the Author):

Although the authors have addressed many concerns raised, the mechanisms involved in CD47-mediated effect is still not clearly defined. Since CD47 is a known ligand for SIRPa, is SIRPa involved in the effect observed? Experiments with SIRPa ko macrophages would strengthen the mechanistic conclusion.

REVIEWERS' COMMENTS:

Reviewer #1 (Remarks to the Author):

The authors have done a thorough job of revision and the manuscript is substantially improved and conclusions better aligned with data.

I still think the conclusions about TGFbeta are overstated, but the authors have toned this down sufficiently that I think it is reasonable for publication.

I agree with the new statement about clodronate liposomes not crossing capillary walls under physiological conditions, what about after injury and vascular leak, and could this affect the results and interpretations?

Overall the manuscript is quite interesting and appropriate for publication in Nature Communications.

Reviewer #2 (Remarks to the Author):

Although the authors have addressed many concerns raised, the mechanisms involved in CD47-mediated effect is still not clearly defined. Since CD47 is a known ligand for SIRPa, is SIRPa involved in the effect observed? Experiments with SIRPa ko macrophages would strengthen the mechanistic conclusion.

-We appreciate this thoughtful comment and have investigated this further using a SIRPa blocking/neutralizing antibody in our burn/tenotomy model. When we block SIRPa during burn/tenotomy, we find that there are no changes in the macrophage phenotype from the M2-like phenotype we saw in the PBS treatment group in our original study. This suggests that it is more than the CD47 activating peptide binding to CD47 and blocking the interaction of SIRPa that is altering the polarization of these cells. We have now included this data in our manuscript (Figure 10e).